# H2A.Z deposition by SWR1C involves multiple ATP-dependent steps

Jiayi Fan[1,2,5], Andrew T. Moreno[3,5], Alexander S. Baier [1,4], Joseph J. Loparo [3] ✉ & Craig L. Peterson [1] ✉

Histone variant H2A.Z is a conserved feature of nucleosomes flanking protein-coding genes. Deposition of H2A.Z requires ATP-dependent replacement of nucleosomal H2A by a chromatin remodeler related to the multi-subunit enzyme, yeast SWR1C. How these enzymes use ATP to promote this nucleosome editing reaction remains unclear. Here we use single-molecule and ensemble methodologies to identify three ATP-dependent phases in the H2A.Z deposition reaction. Real-time analysis of single nucleosome remodeling events reveals an initial priming step that occurs after ATP addition that involves a combination of both transient DNA unwrapping from the nucleosome and histone octamer deformations. Priming is followed by rapid loss of histone H2A, which is subsequently released from the H2A.Z nucleosomal product. Surprisingly, rates of both priming and the release of the H2A/H2B dimer are sensitive to ATP concentration. This complex reaction pathway provides multiple opportunities to regulate timely and accurate deposition of H2A.Z at key genomic locations.

The eukaryotic genome is packaged into the nucleoprotein structure called chromatin, which at its most fundamental level is composed of nucleosomes. The nucleosome consists of two copies of histones H2A, H2B, H3, and H4 wrapped by ~147 bp of DNA[1]. All essential nuclear processes including transcription, DNA replication, and repair require dynamic regulation of chromatin structure by post-translational histone modifications, ATP-dependent chromatin remodeling, and the exchange of canonical histones for their variants[2].

One key histone variant is histone H2A.Z, which replaces canonical H2A in a replication-independent manner at the nucleosomes proximal to gene transcription start sites and transcriptional enhancers, DNA double-stranded breaks, and replication origins[3–5], as such H2A.Z is believed to be a major regulator of gene expression[6], DNA repair[7,8], and replication origin licensing[9], respectively. H2A.Z is evolutionarily conserved from yeast to human, and it is indispensable for fly[10] and mammalian embryonic development[11]. H2A.Z is deposited by

the megadalton complexes of the INO80 subfamily of chromatin remodelers, characterized by SWR1C in yeast[12] and its orthologs Tip60/p400[13] and SRCAP[14] in mammals. Mutations that disrupt these H2A.Z-depositing machines lead to abnormal dosage and localization of H2A.Z on chromatin and can result in a variety of diseases, such as Floating-Harbor Syndrome[15–17], uterine leiomyoma[18], and cancer[19–21].

Chromatin remodeling enzymes have been separated into four distinct subfamilies−SWI/SNF, ISWI, CHD, and INO80[2]. Most chromatin remodelers, including SWR1C, initiate their remodeling activity by binding to superhelical location 2 (SHL 2) of nucleosomal DNA, located 20 base pairs from the center of nucleosomal symmetry, known as the dyad. Prior DNA-histone crosslinking and single-molecule Förster resonance energy transfer (smFRET) experiments have shown that remodelers of the SWI/SNF, ISWI, and CHD subfamilies bind to one DNA strand of SHL2 and induce a ~1−2-nucleotide shift[22–24]. This shift along the presumed tracking strand is then reiterated on the guide

[1]Program in Molecular Medicine, University of Massachusetts Chan Medical School, Worcester, MA 01605, USA. [2]Interdisciplinary Graduate Program, University of Massachusetts Chan Medical School, Worcester, MA 01605, USA. [3]Department of Biological Chemistry and Molecular Pharmacology, Blavatnik Institute, Harvard Medical School, Boston, MA 02115, USA. [4]Medical Scientist Training Program, University of Massachusetts Chan Medical School, Worcester, MA 01605, USA. [5]These authors contributed equally: Jiayi Fan, Andrew T. Moreno. ✉e-mail: Joseph_Loparo@hms.harvard.edu; Craig.Peterson@umassmed.edu

strand, leading to a -1–2 bp DNA translocation toward the dyad upon ATP binding and hydrolysis, synchronizing the ATP cycle to the stepwise translocation of nucleosomal DNA during remodeler sliding. Currently, it is unknown whether the INO80 subfamily performs a similar DNA translocation cycle.

SWR1C stands out among remodelers for utilizing ATP hydrolysis to not slide DNA, but to exchange a nucleosomal H2A/H2B dimer with a H2A.Z/H2B dimer[25]. Previous in vitro studies using bulk assays have shown that SWR1C sequentially deposits two H2A.Z-H2B dimers on a substrate that mimics the +1 nucleosome that is adjacent to the nucleosome-free region (NFR) with an asymmetric preference for NFR-distal exchange[26,27]. However, bulk assays cannot resolve the precise timing of the two exchanges occurring on the same nucleosome or detect reaction intermediates that occur prior, during, and after exchange, and whether they are sensitive to ATP. Furthermore, we do not know how the different phases of the ATP hydrolysis cycle impact the interactions of SWR1C with the nucleosome during dimer exchange.

Here, we address these mechanistic questions of SWR1C remodeling by employing single-base pair and single-nucleosome resolution techniques of site-specific DNA-histone crosslinking and single-molecule Förster Resonance Energy Transfer (smFRET) microscopy, respectively. Using crosslinking, we show that SWR1C does not change the path of DNA on the histone octamer surface during its ATP-dependent deposition of H2A.Z. Instead, single-molecule experiments show that dimer exchange occurs through three distinct phases, two of which are sensitive to ATP concentration. Finally, we use a fluorescence polarization assay to show that ATP hydrolysis dramatically weakens nucleosome binding, consistent with ATP-dependent release of SWR1C following H2A.Z deposition, as suggested by our smFRET studies. Together, these comprehensive assays construct an intricate picture of SWR1C dimer exchange and provide extensive insights on how H2A.Z deposition is regulated on a molecular level.

## Results

### Single-molecule FRET detects preferential and sequential dimer exchange

To probe the dynamics of the H2A.Z deposition reaction in real-time on individual nucleosomes, we developed a smFRET SWR1C dimer exchange assay. A 268 bp, end-positioned nucleosome was reconstituted on a nucleosome positioning sequence harboring a biotin moiety on a 117 bp free DNA linker and an ATTO 647N acceptor fluorophore located on the opposite end. This substrate was engineered to mimic the structure of a promoter-proximal nucleosome that is adjacent to a nucleosome free region[28]. Histone H2A was labeled with a Cy3B donor fluorophore at an engineered cysteine residue, placing the Cy3B and ATTO 647N fluorophores at the appropriate distance to function as a FRET pair (Fig. 1a). Following immobilization of biotinylated nucleosomes on a streptavidin-coated slide, nucleosomes were imaged by total internal reflection fluorescence (TIRF) microscopy. The nucleosomal substrate is similar to that used for ensemble dimer exchange reactions, in which SWR1C activity leads to an ATP-dependent loss of FRET as the Cy3B-labeled dimer is replaced with unlabeled H2A.Z[27]. Since the labeling efficiency of H2A is only ~80%, nucleosome reconstitutions show three clusters of FRET values—nucleosomes with Cy3B on only the linker-distal dimer have a high FRET efficiency (-0.78–0.9), nucleosomes with Cy3B on only the linker-proximal dimer have a low FRET efficiency (-0.4–0.57), and nucleosomes with both labeled dimers have an intermediate FRET value (-0.58–0.77) (see Fig. 1d, right panels; see also Fig. 2a–e for individual, nucleosome trajectories). Consequently, the smFRET assay can report on the timing and efficiency of dimer eviction events from each nucleosome face on individual, surface-immobilized nucleosomes.

The dimer exchange reaction was reconstituted by preincubating SWR1C with nucleosomes and a -2-fold molar excess of H2A.Z/H2B

dimers and the H2A.Z-specific histone chaperone, Chz1 before tethering the substrates in the flow cell[29]. The addition of ATP initiated the exchange reaction, which resulted in the frequent loss of FRET on individual nucleosomal substrates consistent with eviction of the donor-labeled H2A/H2B dimer (Fig. 1b, c). The dynamics of the various FRET states are readily observed by plotting the FRET efficiency as a function of time (Fig. 1d). These kymographs show loss of the initial FRET population (Energy Transfer [E.T.] > 0.5) and the appearance of a low FRET population (E.T. -0.2). In addition, the transition to the low FRET population occurred more rapidly as the ATP concentration was increased (Fig. 1d). Eviction events were defined as persistent loss of FRET from an initial state greater than E.T. = 0.4 to a final FRET state less than E.T. = 0.35. While loss of FRET was sometimes observed in the absence of ATP, these events are likely due to photobleaching, and the inclusion of nucleotide greatly increased the frequency of apparent dimer eviction events (ATP $0\,\mu M = 0.036\,min^{-1}$, ATP $0.5\,\mu M = 0.192\,min^{-1}$, ATP $5\,\mu M = 0.326\,min^{-1}$, ATP $100\,\mu M = 0.48\,min^{-1}$; Supplementary Table 1).

Given that there are two H2A/H2B dimers per nucleosome, and the labeling of H2A was sub-stoichiometric, five primary types of eviction events are anticipated (Fig. 2a–e). For nucleosomes harboring two labeled H2A/H2B dimers (Fig. 2a, b), there was a preferential, ATP-dependent decrease to a low FRET value, consistent with asymmetric exchange of the linker-distal dimer, as previously observed by an ensemble FRET assay (Fig. 2a, f)[27]. Importantly, 40% of these nucleosomes also showed a second decrease in FRET, consistent with a stepwise exchange of the two H2A/H2B dimers (Fig. 2a, b, f and Supplementary Fig. 1). We also observed a population of nucleosomes where the two dimers appeared to be evicted in a single step, though these events may also represent nucleosomes harboring only a single, labeled dimer (Fig. 2e, f).

### smFRET detects multiple ATP-dependent steps during H2A.Z deposition

Analysis of individual FRET trajectories showed that the H2A.Z deposition reaction had three distinct phases (Fig. 3a). Following addition of ATP, nucleosomes exhibited a "priming" period prior to a stable loss of FRET. This phase is unlikely to reflect a lag in SWR1C nucleosome binding, as reactions were preincubated with SWR1C at concentrations promoting single turnover kinetics. Priming is ATP-sensitive, as the duration of the priming phase ($t_{prime}$) decreased with increasing ATP concentration (Fig. 3b). For instance, in the presence of 0.5 μM ATP the lifetime of the priming phase was 92 (82.8–102.9 95% C.I.) seconds and decreased to 55 (51.0–59.6 95% C.I.) seconds with 100 μM ATP (Fig. 3b and Supplementary Table 1). Following the priming phase, the loss of FRET occurred rapidly. The length of time for eviction ($t_{evict}$) was -2–3 s, and it was largely insensitive to ATP concentration (Fig. 3c, e). After rapid eviction, the H2A-H2B dimer remained associated with the H2A.Z nucleosome, with complete loss from the immobilized nucleosome occurring over a longer period ($t_{release}$). Surprisingly, the half-life of $t_{release}$ decreased with increasing ATP concentration, 219 (204.5–240 95% C.I.) seconds at 0.5 μM ATP and 96 (68.0–120.5 95% C.I.) seconds at 100 μM ATP, indicating that the release of H2A/H2B from the final product is an ATP-dependent step (Fig. 3d, Supplementary Fig. 1, and Supplementary Table 1).

Further analysis of the initial, priming phase of the H2A.Z deposition reaction revealed a large number of reversible transitions between high and low FRET states (Fig. 4a, b). Fluctuations in FRET were scored as reversible if the transitions initiated from an E.T. state > 0.6 to an E.T. state <0.45 and returning to the initial state. Very few FRET fluctuations were observed in the absence of SWR1C (Supplementary Table 3), and the transitions increased in frequency and duration after ATP addition to SWR1C reactions (Supplementary Table 1, Supplementary Table 3, and Fig. 4b, d). In the absence of

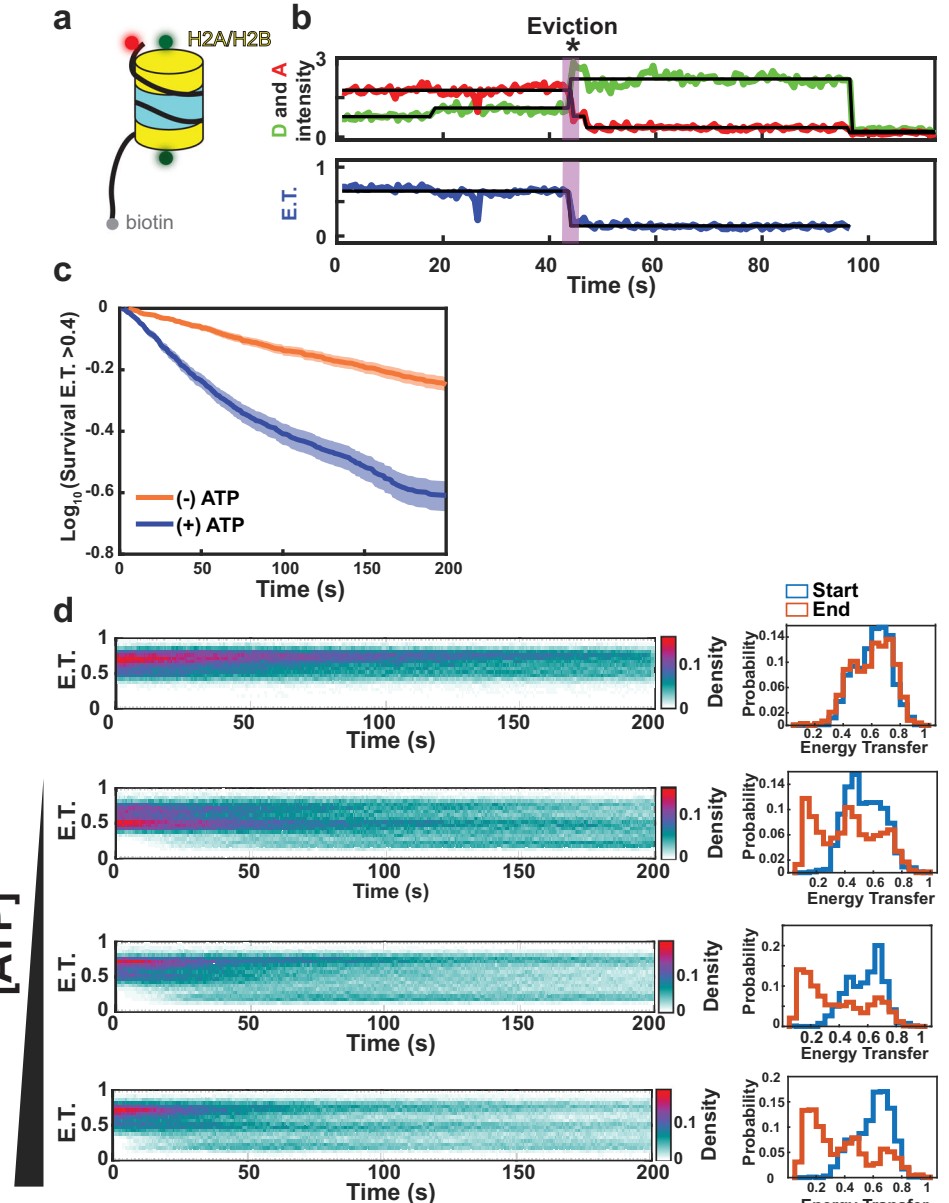

**Fig. 1 | SWR1C eviction of H2A from the nucleosome is ATP-dependent.**
**a** Schematic of nucleosome substrate for smFRET studies. Mononucleosomes contained 4 and 117 bp DNA linkers, with an ATTO 647N fluorophore positioned on the end of the short linker and a biotin group on the long linker. Histone H2A was labeled at the C-terminal domain with a Cy3B fluorophore. **b** Example trajectory. The Cy3B donor fluorophore was excited, and donor emission (green) and ATTO 647N acceptor emission (red) were recorded (top panel) and used to calculate energy transfer efficiency (blue, bottom panel). The eviction of H2A is marked by an *. **c** Energy transfer (>0.4) survival kinetics (Kaplan–Meier estimate) for 0 μM ATP (N = 962 nucleosomes from 9 replicates) (orange) and 100 μM ATP

($N$ = 939 nucleosomes from 8 replicates) (blue). The $x$-axis indicates dwell time for the fraction of nucleosomes remaining in the high E.T. state, the solid line represents the fit from Kaplan–Meier estimate. Shaded areas, 95% confidence intervals. **d** Right-side: Time-resolved energy transfer histograms for tethered nucleosomes with increasing ATP concentrations (0, 0.5, 5, 100 μM) right-side: histogram of the FRET distribution at the start (blue) and end (red) of the trajectory. Note that data obtained for the second set of panels from the top used a nucleosome substrate that was under-labeled, leading to a starting distribution with a lower average FRET value. Source data has been provided for panels (**c**) and (**d**).

ATP, we observed reversible FRET transitions in 55/491 trajectories (1.2 min⁻¹), with an average dwell time of 1.49 ± 2.32 s, while in the presence of ATP, we observed 100/487 trajectories (2.1 min⁻¹) having FRET changes with a dwell time of 3.4 ± 5.8 s (Supplementary Table 3). Given their reversible nature, these FRET transitions likely correspond to transient nucleosome unwrapping events (Fig. 4c). Such unwrapping events are consistent with our previous ensemble FRET data as well as recent three-color smFRET studies from Wu and colleagues[27,30].

To confirm that the stable, ATP-dependent loss of FRET signal is due to H2A/H2B dimer exchange and not stable DNA

unwrapping, smFRET analyses were performed with a nucleosomal substrate harboring unlabeled DNA and the FRET fluorophores located on histones H2A and H3 (Supplementary Fig. 2b). Following the addition of SWR1C and ATP, dimer exchange also exhibited three distinct phases (Supplementary Fig. 2c–e; Supplementary Table 2), including the rapid loss of FRET that follows a lag, or priming period. In the absence of ATP, few FRET fluctuations were observed during the priming period, but surprisingly there was a significant increase in reversible FRET fluctuations during the priming period in the presence of both SWR1C and ATP (Supplementary Fig. 3 and Supplementary

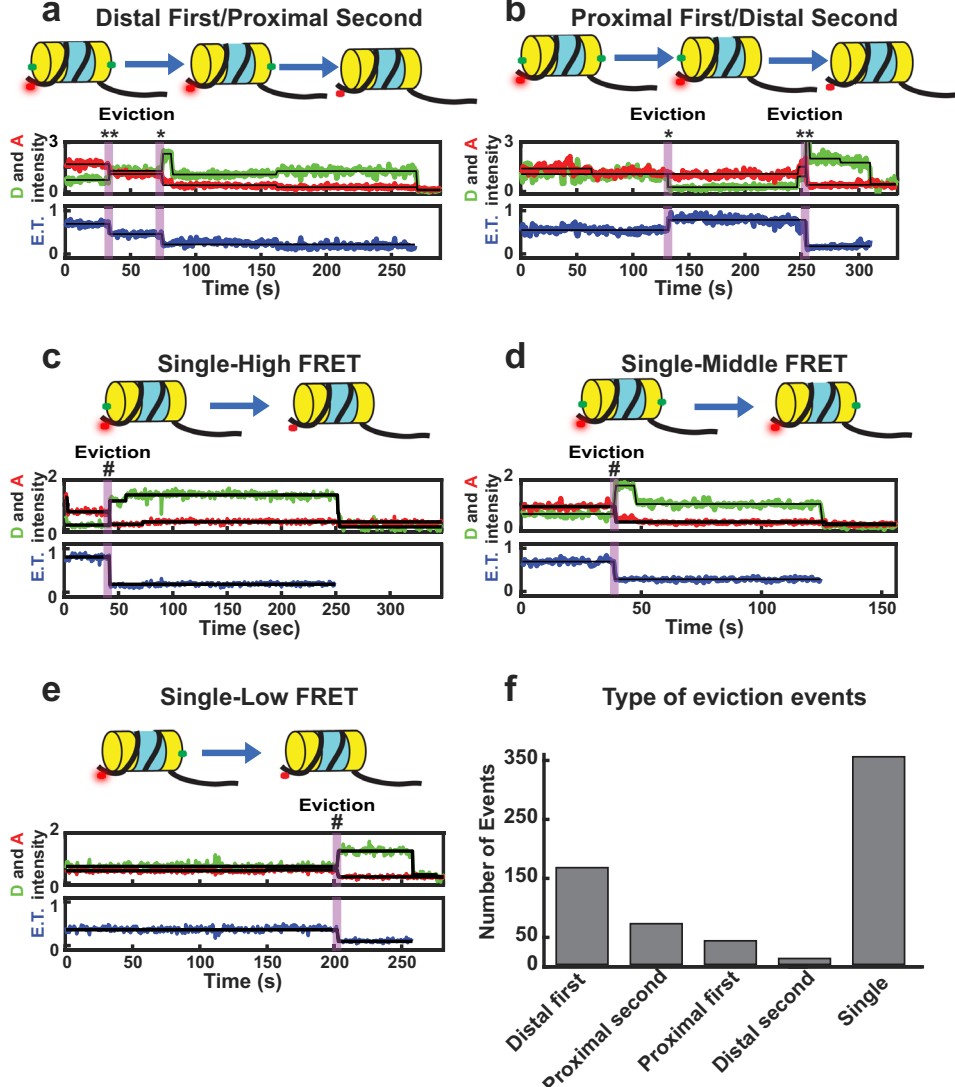

**Fig. 2 | SWR1C evicts distal and proximal H2A from the nucleosome.**
**a−e** Example trajectories. The Cy3B donor fluorophore was excited, and donor emission (green) and ATTO 647N acceptor emission (red) were recorded (top panel) and used to calculate energy transfer efficiency (blue, bottom panel). **a** Example trajectory of SWR1C evicting distal H2A (**) from the nucleosome followed by the eviction of proximal H2A (*). **b** Example trajectory of SWR1C evicting proximal H2A (*) from the nucleosome followed by the eviction of distal H2A (**).

**c** Example trajectory of SWR1C eviction of H2A (#) from a nucleosome in a high energy transfer state. **d** Example trajectory of SWR1C eviction of H2A (#) from a nucleosome in a medium energy transfer state. **e** Example trajectory of SWR1C eviction of H2A (#) from the nucleosome in a low energy transfer state. **f** Observed events for each type of eviction (**a−e**) in the presence of 100 μM ATP ($N = 939$ from 8 replicates). Single eviction represents the summation of events **c−e**.

Table 3). These data suggest that the priming phase may involve not only transient DNA unwrapping events, but also the transient deformations of the histone dimer-tetramer interface. These events are then followed by a rapid eviction of the nucleosomal H2A/H2B dimer.

To directly monitor the deposition of H2A.Z/H2B dimers in real-time, nucleosomes were reconstituted with ATTO 647N-labeled DNA and unlabeled histone octamers, while free H2A.Z/H2B dimers were labeled on H2A.Z with Cy3B (Supplementary Fig. 4). Nucleosomes were first incubated with SWR1C and immobilized on a streptavidin-coated slide. The deposition reaction was then initiated with the addition of Cy3B-H2A.Z/H2B dimers and ATP. Trajectories of Cy3B-H2A.Z fluorescence at individual nucleosomes showed association of single or at times two or more Cy3B-H2A.Z/H2B dimers (Supplementary Figs. 5 and 6). This simultaneous association of multiple dimers may reflect non-specific interaction with the nucleosome/DNA or the ability of SWR1C to interact with more than one dimer on the nucleosome. Control regions of

interest that lacked nucleosomes showed far fewer Cy3B-H2A.Z binding events (Supplementary Fig. 4c) confirming that most binding is nucleosome specific. Nearly all nucleosomes ($N = 307$) exhibited binding (89%) of at least one Cy3B-H2A.Z dimer over the course of 5 min of data acquisition (Supplementary Fig. 4). Cy3B-H2A.Z/H2B dimers rapidly colocalized with immobilized nucleosomes. For instance, we observed Cy3B-H2A.Z/H2B dimers binding to 50% of the nucleosomes within 10 s (Supplementary Fig. 4d). The interaction between a Cy3B-H2A.Z/H2B dimer and SWR1C-nucleosome was relatively stable with a mean lifetime of 34 s ($N = 1,015$), although some individual Cy3B-H2A.Z/H2B dimers remained associated for hundreds of seconds (Supplementary Figs. 4 and 5, and Supplementary Table 4). However, as many of the tethered nucleosomes showed multiple dimers of Cy3B-H2A.Z/H2B simultaneously associated (Supplementary Figs. 5 and 6), the average lifetime of individual dimers may be shorter. After colocalization, a subset of trajectories exhibited an increase in FRET ($N = 29$), likely due to successful deposition of H2A.Z (Supplementary Figs. 4b and 5 and Supplementary

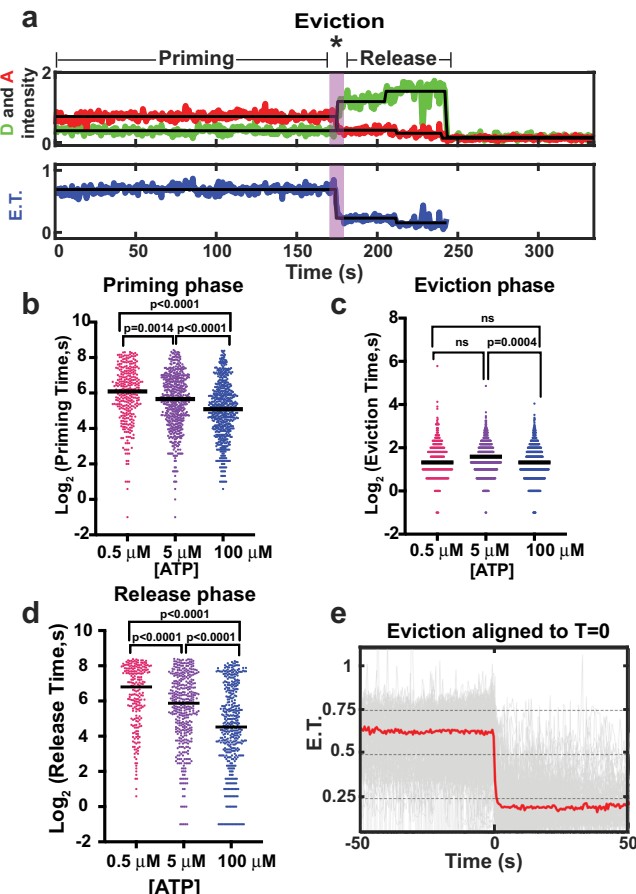

**Fig. 3 | The duration of the priming and release phase is dependent on ATP concentration. a** Example trajectory highlighting priming, eviction, and release intervals. The Cy3B donor fluorophore was excited, and donor emission (green) and ATTO 647 N acceptor emission (red) were recorded (top panel) and used to calculate energy transfer efficiency (blue, bottom panel). The * indicates eviction of H2A. **b** Distribution of the priming phase duration for the indicated ATP concentration, 100 µM ATP $N = 635$ observed events for 939 nucleosomes from 8 replicates, 5 µM ATP $N = 567$ observed events for 940 nucleosomes from 8 replicates, 0.5 µM ATP $N = 325$ observed events for 865 nucleosomes from 7 replicates. **c** Distribution of the eviction phase duration for the indicated ATP concentration, 100 µM ATP $N = 635$ observed events for 939 nucleosomes from 8 replicates, 5 µM ATP $N = 567$ observed events for 940 nucleosomes from 8 replicates, 0.5 µM ATP $N = 325$ observed events for 865 nucleosomes from 7 replicates. **e** Distribution of the release phase duration for the indicated ATP concentration, 100 µM ATP $N = 431$ observed events for 939 nucleosomes from 8 replicates, 5 µM ATP $N = 396$ observed events for 940 nucleosomes from 8 replicates, 0.5 µM ATP $N = 227$ observed events for 865 nucleosomes from 7 replicates. In **b–d** black line is the median, **c** ns, not significant, for one-way ANOVA Tukey's multiple comparison test. **e** Alignment of E.T. trajectories with eviction phase centered at zero with 100 µM ATP ($N = 635$ observed events for 939 nucleosomes from 8 replicates), red line is the median. Source data for (**b**) and (**d**) are provided with this manuscript.

Table 4). The low number of deposition events might reflect incomplete labeling of the H2A.Z dimer, photobleaching preceding deposition of H2A.Z, or a confounding impact of nonspecific H2A.Z binding. While most deposition events occur after association of a single Cy3B-H2A.Z/ H2B dimer, some arise after multiple dimers have associated. Importantly, the deposition is not coincident with co-localization and we observe an average lag time of 38 +/− 20 s between the last co-localization event and deposition. In addition, the length of time between ATP injection and when co-localized Cy3B-H2A.Z/H2B was incorporated into the nucleosome was similar to the duration of the priming phase for H2A eviction (Supplementary Figs. 4b and 5, and Supplementary

Table 4), consistent with the anticipated concerted eviction and deposition reaction[26,27].

## Binding of SWR1C to nucleosomes is sensitive to nucleotides

Real-time analyses of the H2A.Z deposition reaction indicated that a replaced H2A/H2B dimer is released from an immobilized nucleosome in an ATP-dependent reaction. Previous work has implicated the Swc5 subunit of SWR1C as a candidate histone chaperone that binds to the released H2A/H2B dimer and facilitates eviction and replacement[31]. One possibility is that the ATP-dependent loss of H2A/H2B from nucleosomes reflects the release of a SWR1C-H2A/H2B complex. This model suggests that the binding of SWR1C to either an H2A- or H2A.Z-containing nucleosome may be regulated by ATP binding or hydrolysis. To test this hypothesis, a ATTO 647N-labeled, 77N4 nucleosome was used in fluorescence polarization assays to quantify the nucleosome binding affinity of SWR1C (Fig. 5). In the absence of nucleotides, SWR1C bound an H2A nucleosome with an apparent $K_d$ of 12 nM ± 2 nM, and binding to an H2A.Z nucleosome occurred with an apparent $K_d$ of 25 nM ± 6.8 nM. Strikingly, addition of ATP led to a large decrease in binding affinity to an H2A-containing nucleosome (31 nM ± 8 nM), and a decrease in binding was also observed with ADP (21 nM ± 4 nM, but not AMP-PNP (14 nM ± 4 nM) (Fig. 5 and Supplementary Fig. 7). Together, these data indicate the strength of SWR1C-nucleosome interactions is modulated during the ATP cycle and hydrolysis may function to release the enzyme following H2A.Z deposition.

## SWR1C does not alter the path of nucleosomal DNA during binding or H2A.Z deposition

The reversible FRET transitions observed during the extended priming phase of the H2A.Z deposition reaction are consistent with transient DNA-histone unwrapping events that were suggested by previous ensemble studies[27]. How SWR1C promotes DNA unwrapping is not clear. Many remodeling enzymes use a cycle of ATP binding and hydrolysis to translocate the ATPase subunit along one strand of DNA in a 3′ to 5′ direction, leading to movement of DNA on the histone octamer surface[2]. One simple possibility is that SWR1C may use such movements to weaken histone-DNA interactions, leading to DNA unwrapping. For other remodelers, this translocation reaction has been monitored using nucleosomal substrates with histones harboring site-specific, photo-activatable crosslinking agents, such as 4-azidophenacyl bromide (APB)[22,32]. For several remodelers, such as Snf2 and Chd1, binding of the enzyme to nucleosomes is sufficient to induce a 1–2 bp movement of one strand of DNA[22,24]. This remodeler-induced perturbation of the DNA path is often sensitive to whether the remodeler is bound to a non-hydrolyzable ATP analog, which is thought to represent an intermediate in the translocation reaction[23,33].

To investigate whether SWR1C can alter the path of nucleosomal DNA, a center-positioned nucleosome (40N40) was reconstituted that harbored an engineered APB-modified cysteine residue on histone H2B-Q59C. One end of the '601' nucleosome positioning sequence was labeled with a Cy5 fluorophore, and the other end was labeled with a FAM fluorophore to visualize DNA-histone crosslinks from each nucleosome face (Fig. 6a). Consistent with previous work[22], APB modification of H2B-Q59C leads to a predominant UV-induced DNA crosslink ±53nt from the nucleosomal dyad (SHL ± 5.5) (Fig. 6b and Supplementary Fig. 8a, b). Addition of the Chd1 remodeler led to a new crosslink at +55 bp, consistent with a 2 bp movement of DNA from the entry/exit point of the nucleosome towards the nucleosomal dyad, as previously shown with this same nucleosomal substrate (Supplementary Fig. 8a)[22]. The Isw2 remodeler also induced a new crosslink at +55 bp, as well as a new crosslink at the opposite nucleosome face at position −55bp (Fig. 6b and Supplementary Fig. 8b). The appearance of new crosslinks on both faces of the nucleosome likely reflects the ability of Isw2 to translocate DNA when bound to either SHL−2.0 or SHL+2.0. In contrast, the addition of SWR1C had no detectable impact

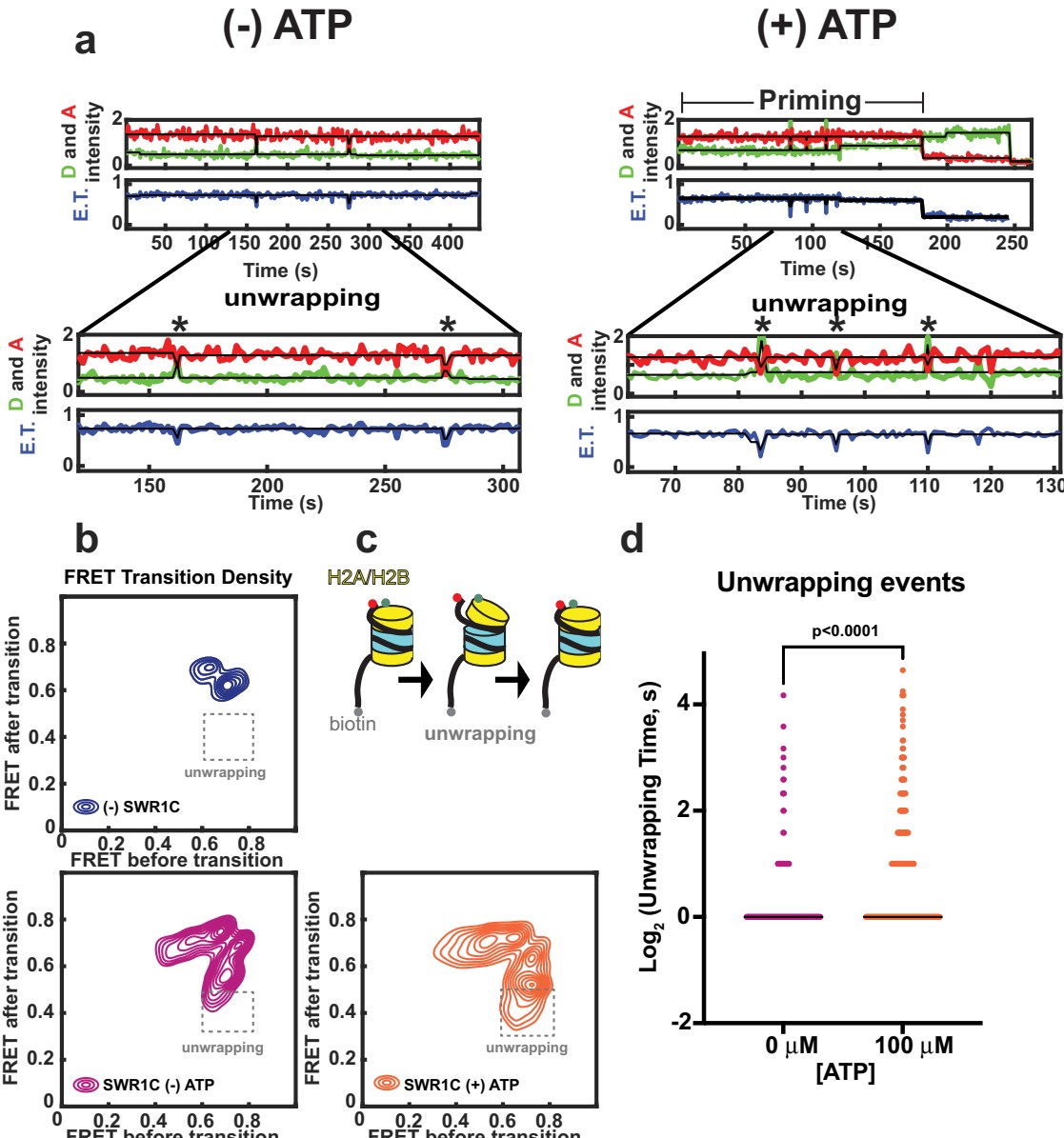

**Fig. 4 | The frequency of SWR1C-induced nucleosome unwrapping increases with the addition of ATP. a** Example trajectory highlighting unwrapping events (*) that occur in the priming phase for 0 µM ATP (left side) and 100 µM ATP. The Cy3B donor fluorophore was excited, and donor emission (green) and ATTO 647N acceptor emission (red) were recorded (top panel) and used to calculate energy transfer efficiency (blue, bottom panel). **b** Priming phase E.T. transition density plot. The intensity of E.T. transitions are normalized to the total observation window, for nucleosome substrate lacking SWR1C and ATP ($N = 73$ nucleosomes with an initial E.T. state > 0.6 from 4 replicates) (blue), and for nucleosome substrate in the presence of SWR1C with 0 µM ATP ($N = 491$ nucleosomes with an initial E.T. state > 0.6 from 9 replicates) (magenta) and 100 µM ATP ($N = 487$ nucleosomes with an initial E.T. state > 0.6 from 8 replicates) (orange). **c** Model of nucleosome unwrapping events, red ball represents Atto647N, and green ball represents Cy3b fluorescent labels. **d** Distribution in unwrapping event dwell times for 0 µM ATP ($N = 491$ nucleosomes with an initial E.T. state > 0.6 from 9 replicates) (magenta) and 100 µM ATP ($N = 487$ nucleosomes with an initial E.T. state > 0.6 from 8 replicates) (orange) for unpaired two-sided student t-test. Source data for panel (**d**) is provided with this manuscript.

on DNA crosslinks at the +53 or −53 position, and the crosslink pattern in the presence of SWR1C was not altered by further addition of ADP or non-hydrolyzable ATP analogs, AMP-PNP or ADP•BeF$_3^-$ (Fig. 6b and Supplementary Fig. 8b).

In our previous study, end-positioned nucleosomal substrates were designed with 77 bp of flanking linker DNA so that it might reflect the asymmetry of a promoter-proximal nucleosome located next to a nucleosome free region (NFR)[27]. On this substrate, SWR1C preferentially exchanges the H2A/H2B dimer that is distal to the long linker, consistent with the pattern of H2A.Z deposition in vivo[34]. Both orientations of an end-positioned nucleosome (77N4 and 4N77) were reconstituted with histones harboring H2B-Q59C, and APB

crosslinking was performed in the absence or presence of SWR1C (Fig. 6c and Supplementary Fig. S8c–e). For both substrates, SWR1C did not induce changes in the crosslinking pattern adjacent to either the distal or proximal H2A/H2B dimer interfaces (Fig. 6c and Supplementary Fig. 8c–e). Furthermore, the addition of H2A.Z/H2B dimer and nucleotides to the SWR1C binding reactions had no impact.

To ensure that SWR1C was active on these APB-modified substrates, H2A.Z/H2B dimers and a low concentration of ATP (3 µM) were added to binding reactions, and a crosslinking time-course was performed (Fig. 6d, e; see also Supplementary Fig. 8e). Addition of ATP led to a time-dependent loss of H2B-DNA crosslinks, consistent with dimer exchange. Furthermore, loss of crosslinks for the linker-distal dimer

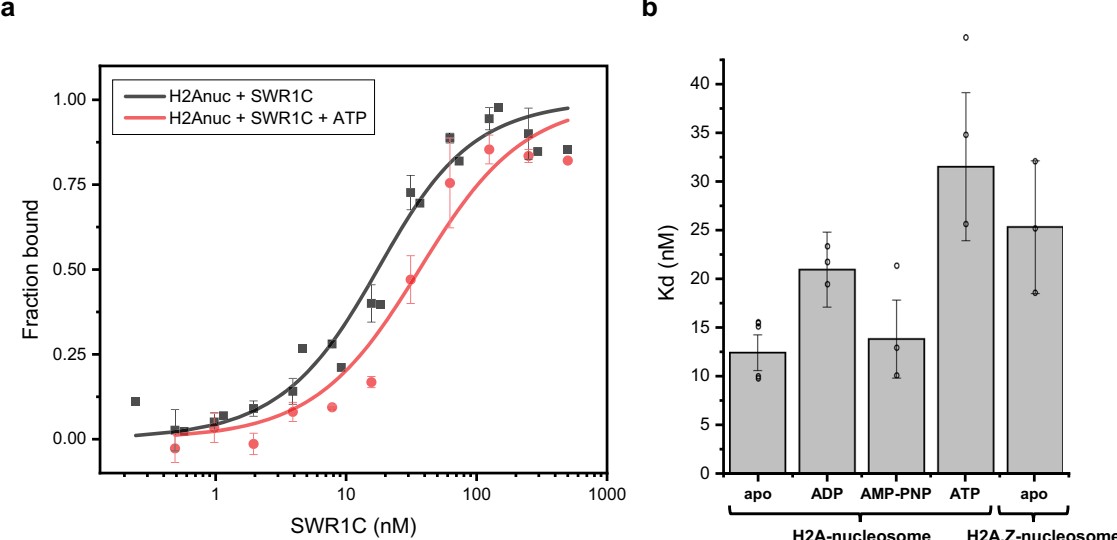

**Fig. 5 | Fluorescence polarization assays show that ATP hydrolysis promotes the release of SWR1C from the H2A-nucleosome. a** Normalized fluorescence polarization plots and fitted curves for titrated SWR1C binding to an H2A nucleosome without nucleotide (black) or in the presence of 1 mM ATP (red). Each curve represents a global fit of 3 replicates with the standard error (+/−SEM) indicated by the error bars. The concentration of SWR1C on the *X*-axis is displayed on a log$_{10}$-scale. **b** The dissociation constant ($K_d$) of SWR1C on nucleosomes in different nucleotide states with 5 replicates in the apo state and 3 replicates for each other condition and the error bars denote +/− SEM. Source data are provided as a Source Data file.

interface occurred more rapidly than the linker-proximal interface, consistent with preferential and asymmetric replacement of the linker-distal dimer (Fig. 6e). Notably, no changes in the crosslinking positions were detected during the dimer exchange reaction. Together, these data suggest that SWR1C may be distinct from other remodelers, and that this remodeler may not induce substantial changes in the path of nucleosomal DNA on the histone octamer surface during H2A.Z deposition[27,35].

## Discussion

SWR1C is unique among the remodeling enzymes characterized to date, as it does not use ATP hydrolysis to alter nucleosome positioning, but rather is dedicated to the replacement of nucleosomal H2A with its variant, H2A.Z[2]. Furthermore, unlike other remodelers, the H2A.Z deposition reaction is kinetically slow even under single turnover conditions, suggesting that there may be a large number of kinetic intermediates in the reaction pathway[27]. Here we use a smFRET approach to identify three ATP-sensitive steps of the H2A.Z deposition reaction−an initial 'priming' step, followed by eviction and replacement of the nucleosomal H2A/H2B dimer, and finally the release of the H2A/H2B dimer from the nucleosomal product (see Fig. 7). The dimer eviction step is quite rapid (~2–3 s), and this timescale is similar to a single cycle of DNA translocation by the ISWI remodeler[36]. In contrast, the initial ATP-dependent priming step is quite slow and is likely to represent the rate-limiting step for the dimer exchange reaction.

### The priming phase during H2A.Z deposition
Following the binding of SWR1C and H2A.Z/H2B dimers to immobilized nucleosomes and the addition of ATP, we observed a long lag phase ($t_{prime}$) prior to the eviction of the Cy3B-labeled H2A/H2B dimer. The duration of this initial priming phase decreased at higher ATP concentrations, indicating that this phase contains one or more ATP-dependent steps. A previous analysis of the ACF remodeler also identified ATP-dependent steps prior to DNA translocation[36]. However, in the case of ACF, a major component of this lag period was the ATP-sensitive binding of ACF to nucleosomes. Binding of SWR1C is unlikely to be influenced by ATP in our studies, as we first pre-incubated an excess of SWR1C with nucleosomes prior to their immobilization and

ATP addition. Furthermore, in contrast to ACF, our nucleosome binding assays demonstrate that ATP does not enhance the binding of SWR1C, but rather significantly weakens it. Thus, if the priming phase was due to unstable SWR1C binding, then increasing ATP levels is expected to lengthen the duration of this step, rather than shorten. One hallmark of the initial priming phase of the H2A.Z deposition reaction is a series of FRET fluctuations that increase in frequency following ATP addition. These changes correlate well with our previous fluorescence correlation spectroscopy FRET (FCS-FRET) studies where we found that the binding of SWR1C to nucleosomes increased the dynamics of DNA-histone interactions at the nucleosomal edge by several orders of magnitude[27]. These nucleosomal dynamics were then further enhanced by addition of the ATP analog, ATPγS. In addition, our previous ensemble FRET studies found that SWR1C could induce transient unwrapping of DNA from the nucleosome[27]. Recently, transient unwrapping of nucleosomal DNA by SWR1C was also observed by a distinct smFRET approach[30]. Surprisingly, transient FRET fluctuations were also observed with a nucleosomal substrate harboring the FRET pair on histones H3 and H2A, though in this case, FRET fluctuations required both SWR1C and ATP. These data suggest a model in which the long priming phase reflects the ability of SWR1C to destabilize the nucleosome by inducing both a transient unwrapping of DNA as well as deformations in the H2A/H2B-H3/H4 interface, 'priming' the nucleosome and lowering the energetic barrier to the subsequent eviction and replacement of the H2A/H2B dimer.

### How is the energy of ATP hydrolysis coupled to dimer exchange?
Remodeling enzymes all contain an ATPase subunit that is a member of the large Snf2 family of DNA-stimulated ATPases[2]. The Snf2 family is part of the larger SFII superfamily of DNA/RNA helicases/translocases, and prior studies on monomeric helicases of the SFII family suggest a model whereby a remodeler ATPase cycle leads to a unidirectional, inchworm-like movement of the bi-lobular ATPase along nucleosomal DNA. Since remodelers are anchored to the nucleosomal surface by histone-binding domains, ATPase translocation can lead to movement of DNA on the octamer surface, "pulling in" DNA from the proximal DNA entry site and propagating DNA towards the distal exit site of the

**a**

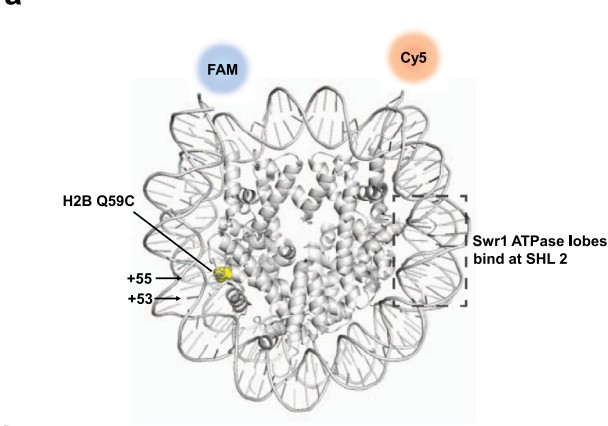

**b**

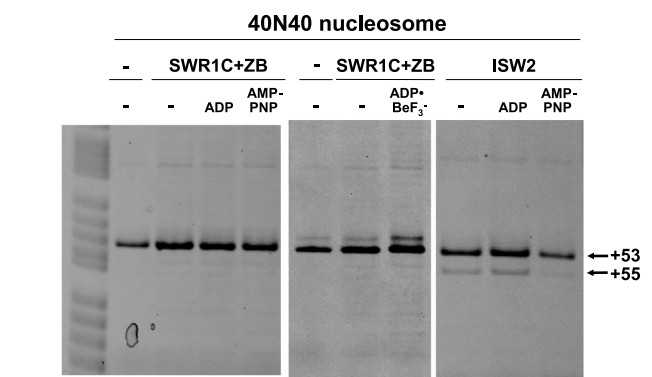

**c**

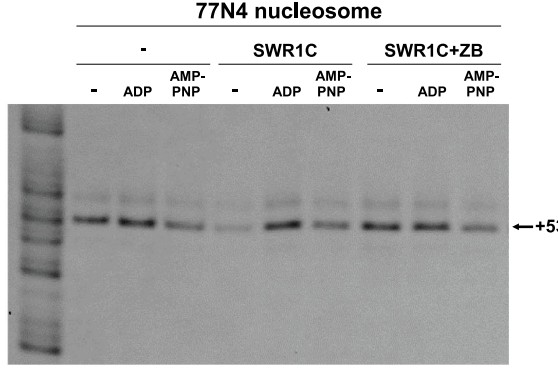

**d**

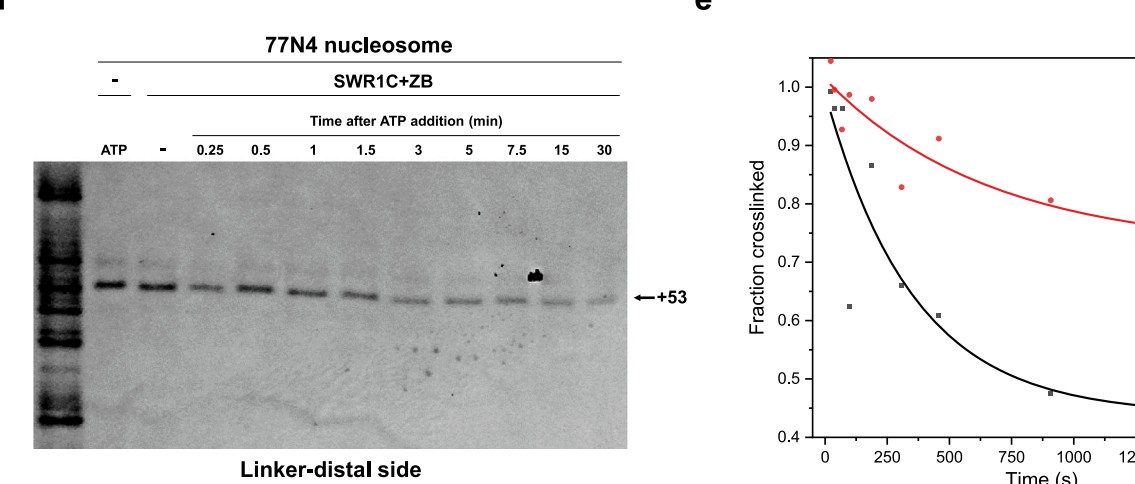

**Fig. 6 | Site-directed DNA-histone mapping shows that SWR1C does not change the path of nucleosomal DNA in any nucleotide state as a part of its ATP-dependent dimer exchange activity. a** DNA-histone mapping schematics on a yeast nucleosome (PDB: 1ID3). APB-labeled H2B-Q59C generates a DNA crosslink at SHL5.5 (+53). **b** ISW2 binding in apo or ADP-bound state induces a 2-nucleotide translocation at SHL5.5 toward the dyad on a center-positioned nucleosome, while SWR1C binding does not, regardless of the presence of nucleotides or H2A.Z/H2B dimer. **c** SWR1C binding does not alter the nucleosomal DNA path at SHL5.5 of an asymmetric nucleosome template. **d** SWR1C dimer exchange is robust in the absence of DNA translocation under 3 μM ATP. **e** Quantification of SWR1C dimer exchange time course shown in (**d**). Crosslinking reactions with SWR1C, Chd1, and Isw2 were repeated with at least two biological replicas, yielding similar results.

nucleosome[2]. Consistent with this model, the introduction of ssDNA gaps between the nucleosomal edge and SHL2 can block the ability of remodelers to mobilize nucleosomes. Furthermore, recent analyses of the Chd1 remodeler suggests that ATP-dependent closure of the ATPase lobes is sufficient to induce a 1 bp translocation step[22].

The general expectation is that SWR1C also employs a DNA translocation mechanism to drive its histone exchange reaction. Consistent with this view, the cryo-EM structure of the SWR1C-nucleosome complex contains a 1 bp bulge of DNA at SHL2[35]. Furthermore, previous studies have shown that 2nt gaps in nucleosomal

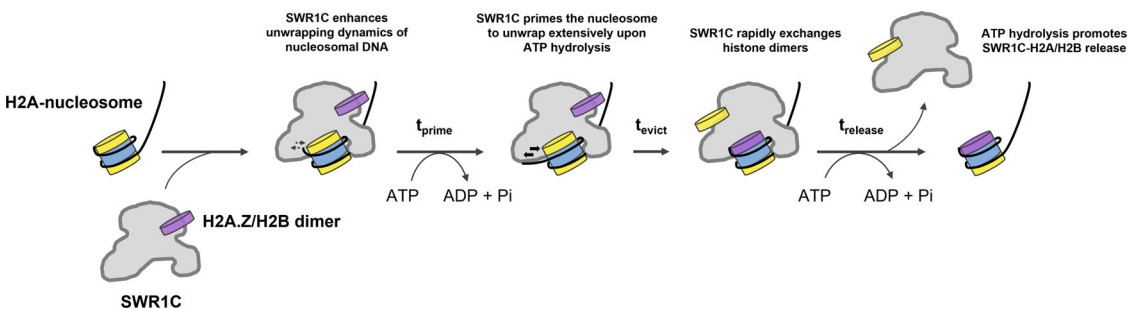

**Fig. 7 | Kinetic model for H2A.Z deposition by SWR1C.** Initially, SWR1C binds to the linker-distal face of an asymmetric H2A-nucleosome, such as the +1 nucleosome adjacent to the NFR. SWR1C binding at SHL2 induces stress and transient dynamics at the nucleosomal DNA edge. While nucleosome-bound, SWR1C hydrolyzes ATP for an extended period of time to prime the nucleosomal substrate for H2A eviction ($t_{prime}$), leading to more extensive unwrapping of nucleosomal DNA and transient

deformations of the histone octamer. Sufficient priming allows SWR1C to rapidly perform its H2A to H2A.Z dimer exchange reaction on the nucleosome ($t_{evict}$). The evicted H2A/H2B dimer remains associated with the SWR1C still bound to the exchanged nucleosome, until the SWR1C-H2A/H2B complex is released from the nucleosome through ATP hydrolysis ($t_{release}$).

DNA can block H2A.Z deposition by SWR1C[37]. However, unlike other remodelers that mobilize nucleosomes, only 2nt ssDNA gaps within the ATPase binding site at SHL2 block dimer exchange (±17 bp to ±22 bp from the nucleosomal dyad) (Ranjan et al., 2015), suggesting that the Swr1 ATPase lobes may translocate only 1–2 bp of DNA. Alternatively, the gaps near SHL±2 may block SWR1C-induced deformations in DNA that precede a putative translocation step[38]. Our previous ensemble FRET studies were unable to detect translocation of DNA at the nucleosomal edge[27], and our histone-DNA crosslinking analyses presented here were also unable to detect changes in the path of nucleosomal DNA between the DNA entry point and SHL+2.0. One possibility is that the crosslinking assay is unable to capture changes in DNA trajectories due to a technical problem. We think this scenario is unlikely, given that the crosslinking assay is able to detect changes in DNA-histone interactions due to the ISW2 and Chd1 remodelers. Alternatively, SWR1C-dependent changes in the path of nucleosomal DNA may be too rapid or dynamic to capture in the assay. We favor a model in which SWR1C distorts DNA at SHL2, as well as the histone octamer, without altering the register of DNA-histone contacts between the entry site and SHL2. Such a distortion would represent a high energy, strained intermediate that promotes subsequent dimer eviction and exchange. In this model, the cryo-EM structure may capture the resolution of this unstable intermediate to a stable product with fully translocated DNA[35].

Unlike remodelers that slide nucleosomes, such as ISW2 and Chd1, SWR1C has a subunit module (Arp6/Swc6/Swc3) that binds to DNA at the linker-distal nucleosomal edge and may prevent the 'pulling' of DNA into the nucleosome[35]. We envision that limited translocation by the Swr1 ATPase at SHL2 acts in opposition to this barrier, distorting DNA structure without "pulling" DNA in from the entry site. This may destabilize histone-DNA contacts, leading to the transient unwrapping events that are observed during the priming stage of smFRET trajectories. A productive unwrapping event might involve stabilization of the unwrapped state by other SWR1C subunits, providing an opportunity for rapid dimer eviction and H2A.Z deposition.

## SWR1C-nucleosome binding is regulated by ATP hydrolysis

Perhaps one of the most surprising results from the smFRET analyses was the observation that release of the labeled H2A/H2B dimer from the immobilized nucleosomal product was sensitive to ATP concentration. At saturating levels of ATP (100 μM[26]), the half-life for Cy3B-H2A release was 96 s, and when the ATP concentration was lowered to 500 nM, the lifetime increased to 219 s. Indeed, this last step in the H2A.Z deposition reaction was the most sensitive to ATP levels. One possibility is that the release step describes an ATP-dependent ejection of the dimer from both SWR1C and the H2A.Z

product nucleosome. We favor an alternative model in which this release step reflects the ATP-dependent loss of a SWR1C-H2A/H2B complex from immobilized nucleosomes. In support of this view, we find that the affinity of SWR1C for nucleosomes is dramatically weakened by the addition of ATP. Furthermore, the presence of ADP also weakened the interaction between SWR1C and the nucleosome. In contrast, the addition of the nonhydrolyzable ATP analog, AMP-PNP, had no effect on binding affinity. These data indicate a weakened interaction between SWR1C and the nucleosome either during or post ATP hydrolysis[39,40]. The ATP-dependent release of SWR1C from its H2A.Z nucleosomal product may facilitate re-cycling of the enzyme, promoting either a subsequent round of H2A.Z deposition on the same nucleosome or for transferring the enzyme to a new substrate. Alternatively, it is possible that an ATP-dependent increase in the on and off rates of SWR1C nucleosome binding promotes more effectively sampling of nucleosomes by SWR1C to enhance the rate of dimer exchange and thereby shorten the priming phase. Interestingly, a recent live-cell imaging study also found that the yeast ISW1, ISW2, and Chd1 remodelers employ ATPase activity to promote fast kinetics of target search and chromatin dissociation[41].

## The stepwise asymmetry of the H2A.Z deposition reaction

From yeast to mammals, H2A.Z deposition is often targeted to the nucleosome adjacent to the start site for transcription by RNA polymerase II[4,5]. Often termed the +1 nucleosome, it is inherently asymmetric, with one side flanked by a nucleosome-free region (NFR), and the other side by the +2 nucleosome[28]. Our in vitro nucleosomal substrates mimic the asymmetry of the +1 nucleosome, as it is flanked by a 77–117 bp linker. Furthermore, DNA footprinting and cryo-EM studies have shown that interactions between SWR1C and the long linker DNA orient the ATPase lobes of the Swr1 catalytic subunit to interact with linker-distal SHL+2.0[25,35], and we previously found that this leads to the preferential eviction of the linker-distal H2A/H2B dimer in an ensemble histone exchange reaction[27]. High-resolution, ChIP-exo analyses of nucleosome asymmetry in yeast are consistent with asymmetric dimer exchange and enrichment of H2A.Z on the NFR-distal side of the +1 nucleosome[34].

Two independent lines of evidence reported here are also consistent with asymmetric H2A.Z deposition. First, our histone-DNA crosslinking assays report on the ATP-dependent loss of histone H2A during the exchange reaction. The crosslinking substrates have different fluorophores at each DNA end, so this assay can determine whether a particular dimer is lost preferentially. In this case, the results mirror our previous ensemble FRET assays, supporting a biphasic loss of H2A and a preference for the linker-distal dimer. Secondly, our smFRET substrates contain H2A/H2B dimers which can each contain a

fluorophore, and thus this assay can report on whether SWR1C prefers to exchange dimers located distal or proximal to the long linker DNA. Consistent with ensemble studies, we find that exchange of the linker-distal dimer is more frequent than the linker-proximal dimer. Many of the smFRET trajectories showed two, sequential H2A eviction events, and in these cases there was a clear preference for eviction of the linker-distal dimer in the first event. Interestingly, in these cases the two events were separated by a period that is similar in magnitude to only the initial priming phase, shorter than a period encompassing both priming and release intervals that is observed for single replacement events. This suggests the possibility that the two, sequential events follow a concerted pathway in which SWR1C is committed to both rounds of exchange without full release from the heterotypic H2A.Z/H2A nucleosome. Together, our data have uncovered phases of the H2A.Z dimer exchange reaction and the unexpected requirement of ATP hydrolysis for both priming and release of the nucleosome by SWR1C, providing an understanding of the molecular mechanism of this unique nucleosome-editing reaction.

## Methods

### Nucleosome reconstitution
*S. cerevisiae* (H2A, H2A-K119C, H2A.Z, H2A.Z-K120C, H2B, H2B-Q59C, H2B-S93C), *X. laevis* (H3.1, H4, H3.1-G33C), and *H. sapiens* (H3.2 C110A) histones were expressed in *E. coli* Rosetta 2(DE3)pLysS cells (except H4 which used Rosetta 2) and purified as previously described[42]. The point substitutions on H2A, H2B, and H3 were generated using site-directed mutagenesis. The purified histones were reconstituted into yeast H2A/H2B-Xenopus/human H3/H4 octamers and yeast H2A.Z/H2B dimers[42]. For smFRET, H2A-K119C and H2A.Z-K120C histones were labeled with Cy3B-maleimide (Cytiva #PA63131) and xH3.1-G33C with ATTO 647N, prior to octamer and dimer reconstitution, respectively, as described previously[43]. The purified octamers and dimers were diluted 1:1 with freeze buffer (10 mM Tris-HCl, pH 7.4, 2 M NaCl, 40% glycerol, 1 mM DTT), flash frozen in aliquots, and stored at −80 °C for nucleosome reconstitution and downstream assays. DNA fragments containing an end- or center-positioned 601 nucleosome positioning sequence were generated via Taq PCR amplification in ThermoPol Buffer and purified using a DNA Clean & Concentrator kit (Zymo #D4004)[27]. Labeled DNA were prepared with 5′-modified primers (IDT) for the following experiments: FAM/Cy3 or FAM/Cy5 for DNA-histone mapping on the 77N4 or 40N40/4N77 template, respectively, biotin-TEG/ATTO 647N for smFRET, and ATTO 647N for fluorescence polarization. Oligonucleotides are shown in Supplemental Table 5. Mononucleosomes were reconstituted at 300–1500 nM concentration by salt gradient dialysis as previously described[27,42]. For DNA-histone mapping, nucleosomes were reconstituted without reducing agent in the dialysis buffer using octamers with no cysteine except at the desired crosslinking site. Note that our previous work demonstrated that the rates of H2A.Z deposition by SWR1C are identical on yeast/Xenopus hybrid nucleosomes and yeast nucleosomes[27].

### Purification of yeast chromatin remodelers
SWR1C and ISW2 were purified from a FLAG-tagged Swr1 or Isw2 yeast strain, respectively, similar to methods described previously[27,44] with further modifications: The harvested yeast pellet was pushed through a 60 mL syringe into liquid nitrogen to generate fine frozen noodles. The noodles were gently crushed with pestle and fully lysed using a PM 100 cryomill (Retsch) with 7 × 1 min cycles at 400 rpm. Following the final B-0.1 buffer wash (25 mM HEPES-KOH, pH 7.6, 100 mM KCl, 1 mM EDTA, 2 mM MgCl₂, 10 mM β-glycerophosphate, 1 mM Na-butyrate, 0.5 mM NaF, 10% glycerol, 0.05% Tween-20, 1 mM DTT, protease inhibitors (PIs): 0.05 µM aprotinin, 1 mM benzamidine, 3 µM chymostatin, 4 µM leupeptin, 3 µM pepstatin A, 1 mM PMSF), the remodeler-bound resin was resuspended in B-0.1, transferred into 1.5 mL Eppendorf tubes, centrifuged at 3000 × *g* for 4 min at 4 °C, and aspirated. The resin was

incubated with 1 mL of 0.5 mg/mL 3× FLAG peptide (Sigma-Aldrich) in B-0.1 and rotated at 4 °C for 1 h to elute the tagged remodeler. The elution was collected by centrifugation and repeated. The combined elution was concentrated with a 50 kDa cutoff Amicon Ultra-0.5 mL centrifugal filter (Millipore #UFC5003) by spinning at 14,000 × *g* at 4 °C, and then flash frozen in aliquots and stored at −80 °C.

Chd1 was purified using tandem affinity purification (TAP) from a TAP-tagged Chd1 yeast strain[45]. 6 L of the tagged strain was grown in 2% glucose YPD media to an OD of ~4 before harvesting by centrifugation at 3000 × *g* for 15 min at 4 °C. The pellet was washed by resuspension and centrifugation with cold water, and then with E buffer (20 mM HEPES-NaOH, pH 7.4, 350 mM NaCl, 10% glycerol, 0.1% Tween-20, 1 mM DTT, PIs). The washed pellet was passed through a 60 mL syringe into liquid nitrogen to generate frozen noodles, which were lysed by cryomilling. The powder lysate was dissolved in an equal volume of E buffer and clarified by ultracentrifugation at 142,000 × *g* for 2 h at 4 °C. The supernatant was incubated on a nutator with 300 µL IgG resin (Cytiva # 17096902) and fresh PIs added for 2.5 h at 4 °C. The slurry was centrifuged at 700 × *g* for 4 min at 4 °C and aspirated. The resin was transferred to an Econo-Pac column (Bio-Rad) and washed three times with 5 mL E buffer without PIs. 300 units TEV protease (in-house prep) in 5 mL E buffer without PIs was added to the resin. The column was capped, wrapped with parafilm, and allowed to nutate overnight at 4 °C. The following day, the supernatant was eluted from the column with fresh PIs added. CaCl₂ was added to a final concentration of 2 mM and the supernatant was added to a disposable column containing 400 µL calmodulin resin (Agilent #214303) pre-equilibrated with E buffer and 2 mM CaCl₂. The column was capped, parafilm-wrapped, and nutated for 2 h at 4 °C. The supernatant was allowed to flow through, and the resin was washed twice with 10 mL E buffer. The resin was incubated in 3 mL E buffer with 10 mM EGTA on the nutator for 10 min at 4 °C to elute the tagged remodeler. The elution was concentrated with a 50 kDa cutoff Amicon Ultra 0.5 mL filter via centrifugation at 14,000 × *g* at 4 °C. The concentrated elution was dialyzed using a 10 kDa cutoff Slide-A-Lyzer MINI dialysis vessel (ThermoFisher #88404) for 3 h at 4 °C into E buffer with 1 mM PMSF and, for RSC only, 50 µM ZnCl₂. The dialyzed sample was flash frozen in aliquots and stored at −80 °C.

All remodeler concentrations were determined via SDS-PAGE using a BSA (NEB) standard titration, followed by SYPRO Ruby (ThermoFisher #S12001) staining and quantification using ImageQuant 1D gel analysis.

### Purification of recombinant histone chaperone Chz1
Chz1 was purified via nickel affinity chromatography as follows: a pQE80L plasmid containing Chz1 harboring a N-terminus hexa-histidine tag was transformed into E. coli Rosetta 2(DE3)pLysS cells. 3 L of transformed cells were grown in 2xYT media to OD 0.7 and induced with 0.8 mM IPTG overnight at 18 °C. The cells were harvested the next day by centrifugation at 3000 × *g* for 15 min at 4 °C. The pellets were resuspended in 50 mL wash buffer (20 mM Tris-HCl, pH 8.0, 500 mM NaCl, 5 mM BME, 1 mM PMSF) with 10 mM imidazole, sonicated, and clarified by centrifugation at 23,000 × *g* for 20 min at 4 °C. The supernatant incubated with 150 µL Ni-NTA affinity resin (QIAGEN #30210) on the nutator at 4 °C for 3 h. The supernatant was allowed to flow through, and the resin was washed with 5 mL wash buffer containing 10 mM imidazole, 5 mL wash buffer with 40 mM imidazole, and eluted with three rounds of 1.5 mL wash buffer with 200 mM imidazole. The fractions were checked via SDS-PAGE and the cleanest Chz1-containing fraction was concentrated using a 10 kDa cutoff Vivaspin 6 concentrator (Sartorius #VS0601) at 3000 × *g* at 4 °C. The concentrated Chz1 was dialyzed into storage buffer (20 mM HEPES-NaOH, pH 7.5, 150 mM NaCl, 1 mM TCEP), flash frozen in aliquots, and stored at −80 °C. Chz1 concentration was determined to be ~100 µM by UV absorbance at 276 nm.

## Site-directed DNA-histone mapping

Site-directed crosslinking to map DNA-histone contacts was performed as previously described[46,47]. 1–1.5 μM H2B-Q59C nucleosomes with FAM/Cy3- or FAM/Cy5-conjugated DNA were labeled with 200–400 μM 4-azidophenacyl bromide (APB) prepared fresh as a 80 mM stock dissolved in N,N-dimethylformamide (DMF). The nucleosomes were labeled for 3 h in the dark at room temperature before being quenched with 5 mM DTT and stored on ice. The crosslinking reactions were prepared in 50 μL volume with 100–150 nM APB-labeled nucleosomes, 300 nM remodeler, 450 nM H2A.Z-H2B dimer for the SWR1C experiments, and 1 mM nucleotide in reaction buffer (For SWR1C: 25 mM HEPES-KOH, pH 7.6, 70 mM KCl, 0.2 mM EDTA, 5 mM MgCl$_2$, 5% glycerol, 1 mM DTT, 0.1 mg/mL BSA; for Chd1/ISW2/RSC: 20 mM Tris-HCl, pH 7.5, 50 mM KCl, 5 mM MgCl$_2$, 5% sucrose, 1 mM DTT, 0.1 mg/mL BSA). ADP stock was prepared by incubating a 100 mM stock to a final concentration of 44 mM with 1 M (18%) glucose, 0.1 U/ul hexokinase, and 5 mM MgCl$_2$ for 20 min at room temperature prior to use. ADP•BeF$_3^-$ was prepared by adding to a final concentration of 1 mM ADP, 6 mM NaF, 1.2 mM BeCl$_2$, and 2.5 mM MgCl$_2$ to the reaction buffer. The reactions were incubated at room temperature for 15–30 min. The reactions were transferred to a 96-well UV-transparent plate (Corning) and irradiated with a UV TransIlluminator (VWR) at 302 nm for 15 s to crosslink. The reactions were transferred back into Eppendorf tubes and mixed with 100 μL quench buffer (reaction buffer with 5 mM EDTA, 5 mM DTT) and 150 μL post-irradiation buffer (20 mM Tris-HCl, pH 8, 50 mM NaCl, 0.2% SDS). The SWR1C dimer exchange reaction was initiated by adding 3 μM ATP and immediately UV-crosslinked and mixed with quench and post-irradiation buffer at the appropriate time points. The samples were vortexed and incubated for 20 min at 70 °C. The incubated samples were added 300 μL 5:1 phenol:chloroform, vortexed, and centrifuged for 2 min at 16,100 × g at room temperature. ~250 μL of the top aqueous layer was removed from each sample without disturbing the aqueous-organic interface and 280 μL of wash buffer (1 M Tris-HCl, pH 8, 1% SDS) was added to the sample, which was then vortexed and centrifuged. This wash was repeated three more times. Crosslinked DNA was precipitated by adding 1.5 μL 10 mg/mL sonicated salmon sperm DNA (Agilent #201190), 33 μL 3 M sodium acetate, pH 5.2, and 750 μL 100% EtOH. The samples were vortexed and incubated on ice at 4 °C overnight. The next day, the precipitated DNA was pelleted at 16,100 × g for 30 min at 4 °C. The supernatant was carefully removed with a pipet. The pellet was washed with 750 μL 70% EtOH and centrifuged at 16,100 × g for 5 min at 4 °C twice. The pellet was air-dried by inverting the opened Eppendorf tube for at least 1 h in the dark. The dried pellet was resuspended with 100 μL resuspension buffer (20 mM ammonium acetate, 0.1 mM EDTA, 2% SDS. The sample was vortexed for 30 s and centrifuged at 16,100 × g for 10 min at room temperature. The supernatant was transferred to another Eppendorf tube and incubated for 2 min at 90 °C. The sample was pulsed, added 5 μL 2 M NaOH, vortexed, and incubate for 45 min at 90 °C to cleave the crosslinked DNA. The sample was pulsed to collect any condensate and added 105 μL 20 mM Tris-HCl, pH 8 and 6 μL 2 M HCl. The sample was then vortexed, added 2 μL 1 M MgCl$_2$ and 480 μL 100% EtOH, vortexed again, and left to precipitate overnight at −20 °C. The following day, the sample was pelleted at 16,100 g for 30 min at 4 °C. The supernatant was carefully removed with a pipet. The pellet was washed with 750 μL 70% EtOH and centrifuged for at 16,100 × g for 5 min at 4 °C twice. The pellet was air-dried by inverting the opened Eppendorf tube for at least 1 h in the dark. The dried pellet was resuspended with 4 μl of 90% formamide, 10 mM NaOH, 1 mM EDTA, and 0.01% bromophenol blue. The sample was vortexed and incubate for 3 min at 90 °C. The heated sample was pulsed and cooled at room temp for 1 min before being loaded onto a denaturing 8% polyacrylamide sequencing gel. A G + A sequencing ladder was used as reference to identify the DNA crosslink location. The sequencing ladder was prepared as previously described (Maxam and Gilbert, 1980) from the corresponding fluorescently labeled DNA used to reconstitute the nucleosome. The gel was run at 65 W for 1.5 h and visualized on a Typhoon Imager by scanning at 473 nm (FAM), 532 nm (Cy3), or 635 nm (Cy5). The dimer exchange crosslinking time course was quantified using ImageQuant 1D gel analysis by normalizing crosslinked band to the uncut DNA band for each time point. Full gel images are provided in Supplementary Fig. 9.

## smFRET imaging and analyses

**Flow cell preparation.** Glass coverslips were placed in coplin jars and cleaned by sonicating for 30 min in methanol After washing with copious amounts of DI H$_2$O, piranha solution (3:1 mixture of sulfuric acid and 30% hydrogen peroxide) was added to the coplin jars and allowed to sit for 1 h. The piranha solution was removed, and coverslips were again washed with DI H$_2$O. The coverslips were then functionalized with a mixture of methoxypolyethylene glycolsuccinimidyl valerate, MW 5000 (mPEG-SVA-5000; Laysan Bio, Inc.) and biotin-methoxypolyethylene glycol-succinimidyl valerate, MW 5000 (biotin-PEG-SVA-5000; Laysan Bio, Inc.) as previously described[48].

Microfluidic chambers were assembled as follows: a diamond-tipped rotary bit was used to drill holes 10 mm apart in a glass microscope slide; a 4.5 mm wide piece of double-sided SecureSeal Adhesive Sheet (Grace Bio-Labs) was placed parallel to one side of holes and across the slide, a second 4.5 mm wide piece of double-sided placed on the other side of the holes making a channel, a piece of functionalized coverslip was then secured to the second side of the adhesive sheet and the edges of the coverslip were sealed with epoxy (Devcon). To make the microfluidic chamber PE20 tubing was inserted into one hole and PE60 tubing into the other (Intramedic), and the tubing was fixed in place with epoxy. The microfluidic cells were stored under vacuum until time of use.

**Preparation of calibration DNA.** The calibration substrate used for channel alignment consists of Cy5 and Cy3 labeled 60 bp duplex. The substrate was made by mixing 10 μM of oligos (IDT) oAM200, oTG415, and oTG416 (Supplementary Table 4) in 100 μL of 20 mM Tris, 300 mM NaCl, 1 mM EDTA pH 8.0. The sample was then placed in a 2 L beaker containing water heated to 90 °C and allowed to cool overnight to room temperature.

**smFRET imaging.** Imaging of single-molecule nucleosomes was achieved using a through-objective TRIF microscope configured around an inverted Olympus IX-71 microscope. Each laser beam 532 nm (Coherent Sapphire 532) and 641 nm (Cube 641) were first expanded and then combined using dichroic mirrors. The combined beams were expanded again and focused onto the rear focal plane of an oil-immersion objective (Olympus UPlanSApo, 100 3; NA, 1.40). To achieve TRIF illumination the focusing lens was manually translated in the vertical plane. A multipass dichroic mirror was used to separate emission from excitation light. The emission light was then sent across a StopLine 488/532/635 notch filter (Semrock) to further reduce excitation light. A home-built beamsplitter[48] was used to separate emission from Cy3B and ATTO 647N and the then imaged on separate halves of EMCCD camera (Hamamatsu, ImageEM 9100-13) operating at maximum EM gain. The focus was adjusted manually, and the sample was positioned on the microscope using an automated microstage (Mad City Labs).

For calibration data the flow cell chamber was incubated with 35 μL 0.71 mg/mL streptavidin (25 μL 1 mg/mL streptavidin in 1× PBS diluted with 10 μL of 10 mM Tris-HCl, pH 7.4, 1 M NaCl) for 5 min. The calibration substrate, 5-prime biotinylated 60 bp dsDNA labeled with Cy5 and Cy3, was diluted to ~30 pM in Tris pH 7.4 (10 mM), NaCl (1 M), EDTA (1 mM), protocatechuic acid (5 mM), protocatechuate 3,4-dioxygenase (0.1 mM), and Trolox (1 mM). The biotinylated, fluorescent DNA substrate was immobilized on a glass coverslip in a microfluidic

chamber. Images were acquired of different fields of view (~120) with 0.5 s simultaneous exposure to 532 and 641 nm lasers using a surface power density of 4 mW/cm² for the 532 nm laser and 2.4 mW/cm² for the 641 nm laser.

The smFRET data was collected at a surface power density of 1.9 mW/cm² for 532 nm and 0.76 mW/cm² for 641 nm. The integration time was 0.5 s per frame and the images were collected continuously at a cycle of 4 frames of 532 nm excitation and 1 frame of 641 nm. The flow cell chamber was incubated with 35 µL 0.71 mg/mL streptavidin (25 µL 1 mg/mL streptavidin in 1× PBS diluted with 10 µL of 10 mM Tris-HCl, pH 7.4, 1 M NaCl) for 15 min. The chamber was passivated with 150 µL smFRET wash buffer (25 mM HEPES-KOH, pH 7.6, 70 mM KCl, 0.2 mM EDTA, 5 mM MgCl₂, 1 mM DTT, 0.2 mg/mL acetylated BSA (Promega), 0.02% NP-40 (Sigma-Aldrich) for 5 min. The chamber was then equilibrated with 150 µL smFRET reaction buffer (20 mM HEPES-KOH, pH 7.6, 56 mM KCl, 0.16 mM EDTA, 4 mM MgCl₂, 10% glycerol, 1 mM DTT, 0.1 mg/mL acetylated BSA, 0.02% NP-40, 1 mM Trolox, 0.8% glucose, 0.24 mg/mL glucose oxidase, 0.24 mg/mL catalase acetylated BSA 0.2 mg/mL, 0.02% NP-40). For dimer eviction reactions, 50 µL of 90 pM biotinTEG-117N4-ATTO 647N, Cy3B-labeled H2A nucleosome, 810 pM biotinTEG-117N0 nucleosome, 10 nM 77N0 nucleosome, 30 nM SWR1C, 70 nM H2A.Z-H2B dimer, and 70 nM Chz1 in reaction buffer was flowed into the chamber and incubated for 5 min. The chamber was then washed with 150 µL wash buffer. The dimer eviction reaction was initiated 20 s after beginning imaging by injecting 50 µL reaction buffer containing the indicated concentration of ATP, 10 nM 77N0 nucleosome, 30 nM SWR1C, 70 nM H2A.Z-H2B dimer, and 70 nM Chz1. The imaging data was collected for a total of 6 min. For dimer deposition reactions, 50 µL of 90 pM biotinTEG-117N4-ATTO 647N nucleosome, 810 pM biotinTEG-117N0 nucleosome, 5 nM 77N0 nucleosome, 25 nM SWR1C, 50 nM Cy3B-labeled H2A.Z-H2B dimer, and 50 nM Chz1 in reaction buffer was first incubated in the chamber and the deposition reaction was initiated with 50 µL reaction buffer containing 200 µM ATP, 10 nM 77N0 nucleosome, 25 nM SWR1C, 50 nM Cy3B-labeled H2A.Z-H2B dimer, and 50 nM Chz1. All smFRET experiments were performed at room temperature.

**Alignment of the donor and acceptor channels.** Using custom MATLAB scripts, an automated spot-detection algorithm was used to identify fluorescent molecules in the 532 and 641 emission channels of each image. Spots were identified by first subtracting local background and identifying particles based on local maximum. For greater precision in location, shape, and amplitude, particles were fit using a 2D Gaussian. For each image spots whose coordinates <6 pixels from surrounding molecules and are within 6 pixels of a spot in the corresponding channel are added to the initial calibration list. The initial calibration list is then refined based on spot diameter and amplitude. A final calibration list is made using the method outlined in the paper from[49]. Briefly, for each image the coordinates $(x_1, y_1)$ of a spot in channel 1 (532 emission) is mapped onto coordinates $(x_2, y_2)$ in channel 2 (641 emission) to identify a matching partner spot using the transformation Eq. (1)

$$x_2 = Ax_1 + By_1 + C \qquad (1)$$

$$y_2 = Dx_1 + Ey_1 + F' \qquad (2)$$

where A–F are fit parameters. Because of systematic various across the field of view, the fit parameters are determined for each spot coordinate $(x_1, y_1)$ by fitting pairs of corresponding points from the initial calibration list that are within a 30×30-pixel box of $(x_1, y_1)$. A spot that maps greater than 2 pixels from all spots in channel 2 is removed form calibration list. This process is repeated for spots in channel 2, mapping coordinates onto channel 1 using the updated calibration list. Channel 2 spot that are within 1 pixel of to its channel 1 partner are

retained in the calibration list. The calibration list is further refined by repeating the above process removing spots that are greater than 1 pixel from spots in the other channel to give the final calibration list that is used to map coordinates between the two channels.

**Detection and quantification of eviction events.** Quantification of single-molecule eviction experiments were performed using the following steps. Spot detection was achieved using an image of the 532 nm channel and 641 nm channel from the start of the movie with the method described above. Molecules within a radius of 6 pixels of other particles were excluded from the analysis to avoid crosstalk. Fluorescent pairs were identified using the calibration list and transformation method described above. Stage drift was estimated in movies using average change in $x$ and $y$ position between successive frames for particles in the 641 nm channel and region of interest (ROI) positions were translated to compensate. Integrated intensities derived from raw images were used for energy transfer efficiency calculations. The integrated intensity of each particle over time was calculated as the summation of a circular ROI of radius 4 pixels centered on the particle. The local background intensity was determined as the median intensity for pixels flanking the ROI. Photobleaching events for ATTO 647N were automatically identified using the built-in MATLAB function findchangepts, which identifies abrupt changes in signal, to first mark changes and aggregate intensities past the change point for all molecules. Next, the aggregated intensity histogram was fit to a Gaussian distribution, and a threshold for ATTO 647N photobleaching was set at two standard deviations above the mean. A similar method was used to define photobleaching threshold (either due to photobleaching or loss of Cy3B by eviction from the nucleosome) for the total Cy3B emission signal, which is the sum of intensity for Cy3B emission and Cy3B excited ATTO 647N emission. The trajectories for each molecule were then automatically truncated based on photobleaching events and if necessary, the point of truncation was manually adjusted.

For each ATP condition the trajectories from a minimum of three independent experiments were combined. The initial energy transfer for each molecule was defined as the median transfer efficiency of the first 10 frames. The trajectories with an initial energy transfer greater than 0.4 were globally fitted to a three-state Hidden Markov Model using the ebFRET MATLAB software package[50] with default parameters. A custom MATLAB script was used to identify eviction events, the duration of the priming, eviction, and release intervals based on the output FRET states from ebFRET. Proximal first eviction events were defined as change a from an initial E.T. state between 0.58 and 0.77 to a E.T. state between 0.4 and 0.57. Distal second eviction events were defined when following a proximal eviction there is a change a from an E.T. state between 0.4 and 0.57 to a E.T. state below 0.35 without loss of Cy3B signal. Distal first eviction events were defined as change a from an initial E.T. state between 0.58 and 0.77 to an E.T. state above 0.77. Proximal second eviction events were defined when following a Distal eviction there is a change a from an E.T. state above 0.78 to an E.T. state below 0.35 without loss of Cy3B signal. Single-High eviction events were defined as change a from an initial E.T. state above 0.77 to an E.T. state below 0.36 without loss of Cy3B signal. Single-Middle eviction events were defined as change a from an initial E.T. state between 0.58 and 0.77 to an E.T. state below 0.35 without loss of Cy3B signal. Single-Low eviction events were defined as change a from an initial E.T. state between 0.4 and 0.57 to an E.T. state below 0.35 without loss of Cy3B signal. The length of time for eviction was determined by first using a 3-point moving median to smooth the E.T. signal and next fitting a baseline to the initial E.T. state and the E.T. state following eviction. The start and stop of the eviction were defined as the point of 5% deviation of smooth E.T. signal from respective baselines. The duration of the priming interval is then defined as the time from the start of the trajectory to the start of eviction. For eviction events that result in a E.T. state below 0.35, the release interval was

demarcated as the length of time between the end of the eviction to the loss of Cy3B signal.

For each type of eviction event the distribution in the duration of priming interval were fit to a single exponential using the expfit function in MATLAB. To calculate the half-life of the release phase for each ATP concentration, survival curves of the time for release (Supplementary Fig. 1d) were constructed using the Kaplan–Meier procedure. Release intervals were classified as right-censored if no donor loss event was observed before the end of the observation interval. Statistical analysis of the priming, eviction, and release interval between the different ATP concentrations (0.5, 5, 100 μM ATP) were performed using an ordinary ANOVA test in Prism Version 9.3.1, $P$ values % 0.05 were considered significant for this analysis. Because of the shape of the priming and release time distributions, the statistical tests were performed on the logarithm base 2 scale to better satisfy the assumption of homoscedasticity. To obtain the kymographs shown in Fig. 1d, energy transfer efficiency trajectories were truncated upon photobleaching and then pooled. Survival curves of the energy transfer (Fig. 1c) were constructed using the Kaplan–Meier procedure. Trajectories were classified as right-censored if the molecule remained in the high E.T. state at the end of the observation interval.

To examine unwrapping events that occurred during the priming interval, trajectories were truncated at the start of eviction. Molecules with an initial energy transfer greater than 0.6 were globally fitted to a three-state Hidden Markov Model using the ebFRET MATLAB software package[50] with default parameters. A custom MATLAB script was used to identify unwrapping events and the duration of the events. Unwrapping events were defined as a transition from an initial state with a median E.T. greater than 0.6 to state with a median E.T. less than 0.45 and a subsequent return to a state with a median E.T. greater than 0.6. Transition density plots were made by first normalizing the intensities of transition to the total number of frames, where intensities indicate the number of transitions between E.T. X and Y per unit time. Intensities that are greater than 0.0005 were plotted using the contour function in MATLAB. Statistical analysis between (−) ATP and (+) ATP (100 μM ATP) were performed using an unpaired t test in Prism Version 9.3.1, $P$ values % 0.05 were considered significant for this analysis.

**Detection and quantification of deposition events.** Analysis of single-molecule deposition of Cy3B-labeled H2A.Z experiments were performed using the following steps. The ROIs for the tethered nucleosomes were determined with an image of the 641 nm channel from the start of the movie using the method described in *Alignment of the Donor and Acceptor Channels*. To examine the non-specific interaction between H2A.Z and the coverslip, control ROIs were picked that were > 6 pixels from tethered nucleosomes. The position of ROIs in the Cy3B channel for tethered nucleosomes and control ROIs were identified using the calibration list and transformation method described in *Alignment of the Donor and Acceptor Channels*. Stage drift was estimated using average change in $x$ and $y$ position between successive frames for particles in the 641 nm channel and ROI positions were translated to compensate. In each of the 532 nm excitation frames, Cy3B-labeled H2A.Z localized spots were detected using the algorithm described in *Alignment of the Donor and Acceptor Channels* with the following modification. The background peak of the Cy3B intensity histogram was fit locally to a Gaussian distribution and a background threshold was calculated as one standard deviation above the mean. Localized H2A.Z spots with an intensity above background threshold were fit to a 2D Gaussian. Integrated intensities of the total Cy3B emission signal (sum of intensity for Cy3B emission and Cy3B excited ATTO 647N emission) was used to identify binding and dissociation of H2A.Z to tethered nucleosomes using the following method. The trajectories for Cy3B were first denoised using a piecewise constant approximation[51]

followed by identification of changepoints using the built-in MATLAB function findchangepts. Binding events were required to satisfy the following conditions: (1) the intensity must increase by 30% at identified changepoint, (2) the intensity must be two standard deviations above the mean Cy3B background signal, (3) the localized H2A.Z must be within 1.25 pixels of the center of the ROI. The dissociation events were defined as changepoints with a greater than 50% decrease in amplitude, which is less than one standard deviations above the mean Cy3B background signal. Trajectories were manually annotated to mark deposition events identified by an anticorrelated increase in Cy3B excited ATTO 647N emission and a decrease in Cy3B emission. To obtain the fraction bound plot (Supplementary Fig. 4d), the initial binding events are sorted by arrival time and the cumulative sum of these events are normalized to the number of surface-tethered nucleosome complexes. The dwell time distribution for H2A.Z colocalized with surface-tethered nucleosomes was fit to a double exponential using the MEMLET MATLAB software package[52]. A bootstrap analysis (1000 iterations) was used to estimate the confidence intervals for the exponential fit.

**Fluorescence polarization nucleosome binding assay**
The fluorescence polarization binding assay was performed as follows: A 2-fold serial dilution of SWR1C-nucleosome binding reactions was prepared in reaction buffer (25 mM HEPES-KOH pH 7.6, 70 mM KCl, 0.2 mM EDTA, 5 mM MgCl$_2$, 1 mM DTT, 0.1 mg/mL BSA) to a final concentration of 0–250 nM SWR1C, 10 nM 77N4-ATTO 647 N nucleosome, 1 mM nucleotide in 20 μL reaction volume. The reactions were transferred onto a 384-well black microplate (PerkinElmer) and incubated for 20 min at room temperature. The fluorescence polarization signal was measured in a Tecan Spark microplate reader using an excitation wavelength of 631 nm and emission wavelength of 686 nm. A binding curve was generated from the SWR1C titration, normalized, and fitted to the quadratic binding equation

$$y = \frac{(x + [S]_t + K_d) - \sqrt{(x + [S]_t + K_d)^2 - 4[S]_t x}}{2[S]_t}$$

where $[S]_t$ is the nucleosome concentration (10 nM), $x$ is the SWR1C concentration, $y$ is the fraction of nucleosome bound, and $K_d$ the dissociation constant to be determined from the fit. At least three replicates were generated for each condition and a global fit on the replicates were performed using OriginLab to calculate the $K_d$.

**Reporting summary**
Further information on research design is available in the Nature Portfolio Reporting Summary linked to this article.

## Data availability
The data that support this study are available from the corresponding authors upon reasonable request. Source data are provided with this paper.

## Code availability
The single-molecule datasets and custom MATLAB analysis code are available from the corresponding authors on request.

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

## Acknowledgements
We thank Nathan Gioacchini (UMMS) for help with DNA sequencing gels, Shinya Watanabe (UMMS) for the expression vector for human H3.2 C110A, and members of the Peterson and Loparo groups for helpful discussions This work was supported by grants from the National Institutes of Health [R35-GM122519] to C.L.P. and [RO1GM115487] to J.J.L.

## Author contributions
Experiments were performed by J.F. and A.T.M.; A.S.B. helped with the fluorescence polarization studies; J.F., A.T.M., C.L.P., and J.J.L. analyzed data and edited the manuscript.

## Competing interests
The authors declare no competing interests.
