## [Peer Review File · Nature Communications]

Editorial Note: This manuscript has been previously reviewed at another journal that is not operating a transparent peer review scheme. This document only contains reviewer comments and rebuttal letters for versions considered at Nature Communications

REVIEWER COMMENTS

Reviewer #1 (Remarks to the Author):

One of the findings of this paper regards what is called the priming phase which is a longer period of time that occurs before there is loss of FRET signal, presumably due to the loss of an H2A-H2B dimer. The authors conclude there is transient unwrapping of DNA from the histone octamer on the linker distal side in conjunction with destabilization of the histone octamer. The data however is not clear in this regard. In this revised version of the manuscript the authors have added new data by place the donor dye on histone H3 instead of on DNA and the acceptor dye is on histone H2A and then finding transient fluctuations like that observed when the donor dye is on DNA. They conclude that there are transient disruptions within the histone octamer that accounts for these transient fluctuations; however, they continue to state the other version with the donor dye on DNA shows that DNA is unwrapped. The original reason for performing this last experiment was to confirm if the fluctuations observed with the DNA-H2A FRET pair is truly a reflection of DNA unwrapping or could be the result of other structural fluctuations. It seems from their most recent experiment the conclusion should be they cannot state the changes in FRET signal with the labeled DNA and H2A are due to DNA unwrapping but instead are likely caused by changes in the histone octamer. If in the situation of the donor on DNA and acceptor on histone H2A, H2A is moving due to transient conformations of the histone octamer that would be sufficient to explain the type of fluctuations observed in this study and not due to the DNA moving in regard to the histone surface. In summary the authors don't have good evidence for DNA unwrapping from the linker distal side of the nucleosome and do not have an explanation for how ATP helps "prime" for eviction of the H2A-H2B dimer.

In the discussion section they also mention they don't think these fluctuations are due to SWR1 binding because SWR1C is not influenced by ATP (see top of page 14). I found this statement to be contradictory to some of the new data that has been added as they now have fluorescence polarization data where they determine the affinity of SWR1 to H2A-nucleosomes with (1 mM) and without ATP, which showed that ATP decreased the affinity of SWR1 for H2A-nucleosomes from 12 to 31 nM and they conclude the strength of the SWR1C-nucleosomes is modulated with ATP. It seems therefore the fluctuations could be due to the dynamic binding and release of SWR1 from nucleosomes which would be influenced by ATP concentration, which further cloud the purpose of the long priming period and could it be merely that it has to keep on going through rounds of binding and releasing before it finally is successful for displacing H2A-H2B. Because of the lack of light shed on the functional aspects of the priming period, the significance of their findings is diminished. A minor point is that although the affinity of SWR1 for H2A nucleosomes was measured w/o nucleotide (Apo) and with ADP, AMP-PNP and ATP, the affinity for H2A.Z nucleosomes was only measured w/o nucleotide. It seems like it would be critical to measure SWR1 affinity for H2A.Z nucleosome with ATP, ADP, and AMP-PNP as well.

distorts DNA only at SHL2 without altering the register of DNA-histone contacts between the entry site and SHL2. They suggest the minor distortion in DNA coupled with distortion of the histone octamer would be the driving forces for dimer exchange. There are however concerns that the data is being misinterpreted. Two key factors need to be kept in mind, namely the strand-specific crosslinking that occurs with modified H2B and DNA translocation occurring on the tracking strand of the ATPase domain. In order for DNA translocation to be effectively observed by this type of photocrosslinking, these two factors have to line up or in other words the strand being tracked by DNA crosslinking has to be the tracking strand of the ATPase domain otherwise you would not expect to detect translocation by this method. CHD1, ISW2 and SWR1C all have the top strand as the tracking strand and SWR1 should be translocating in a 3'-5' direction based on the SWR1-nucleosome cryo-EM structure. The 1-2 bp translocation observed with ISW2 and CHD1 are only observed on the tracking strand from the SHL2 position to the nearest edge of the nucleosome and does not generally transverse across the dyad axis. As you can see from the figure below DNA crosslinking is positioned well to detect the nucleotide induced translocation of only the tracking strand by ISW1 and CHD1, the same is however not true for SWR1C if it did indeed translocate. The DNA crosslinking would have to be on the top strand nearest to the linker distal side of the nucleosome to detect DNA translocation if it occurred by SWR1 but the DNA crosslinking is instead tracking only the bottom strand. For these reasons, the experimental design is not well suited to address the question. If you change the longer linker DNA to the other side of the core nucleosome you would still have the same problem as the tracking strand would now be the bottom strand

I didn't understand their suggestions of how Arp6/Swc6/Swc3 binding near the nucleosomal edge on the long linker side and the small distortion at SHL2 on the opposite side of the dyad axis, some ~90 or more nucleotides from each other, could work together to destabilize histone-DNA interactions without causing DNA to translocate as well. This model just doesn't make sense.

What is missing in this study is an orthogonal approach to track loss of H2A-H2B dimer to validate their changes in FRET signal. Loss of FRET signal could indicate several types of events particularly since there is evidence the histone octamer is being deformed based on their H3-H2A FRET results. The orientation of the acceptor would need to merely change its location

without being dissociated to produce the same change of FRET signal. Because of this lack of further validation it makes their interpretation of these data rather tenuous.

Minor points:

On page 6 the authors allude to the number of dimer eviction events being greater with ATP but don't tell you the number – instead they give you the rate differences. I would like to also know the number of events as stated.

I see in Table S3 that the number of transient FRET fluctuations with and without ATP are only modestly different from each other (152 vs 234). It raises the question if the roughly doubling of the rate with the addition of ATP suggest these maybe intrinsic fluctuations that are modestly modulated by ATP and have no relationship to the process of dimer exchange?

When I look at what is referred to as the release phase in Figure 3, I'm not clear as to the significance of the release phase as not much really seems to be happening based on ET differences especially since ET is already so low and the donor intensity is so high. Is it really an important phase in the dimer exchange process and where is the rationale for it being so?

The reversible steps that occur in the priming seem to be quite similar with and without ATP although there is modest increase with ATP. Not sure how significant these differences truly are.

The new results with labeled H3 and H2A are suppose to show the same 3 phases as seen before but in Supplemental Figure 2B-E I only see evidence for the eviction and priming phases. Where is the missing data? Is the lag period the same as the priming phase? Why is the fluctuations in the priming phase displayed so differently than before, making it difficult to compare and contrast the two? The data is also quite sparse here and begs the question as to the rigor of the experiment.

Where is Supplementary Figure S4 – movie?

There were only 29 deposition events observed out of a total of 1136 events. That number makes one worried about what is being observed and if the system is adequate or optimal for tracking the incorporation of H2A.Z dimers into nucleosomes? I would like another orthogonal dimer incorporation assay before to support these data. The authors state the low number could be due to the incomplete labeling of H2A.Z. My question is if 100% of the H2A.Z dimer was labeled as compared to say ~80% do you think it would make that much of a difference? What was the ATP concentration used for the H2A.Z deposition Sm FRET experiments? I could find it in either the figure legend, main text or methods section?

The authors cite the early work with ACF using smFRET but have failed to reference later studies involving ISW2 where there do observe a similar lag before DNA is moved through the nucleosome that is not dependent on the kinetics of ISW2 and could be considered to resemble the data reported here (Deindl et al Cell 2013).

Reviewer #2 (Remarks to the Author):

The authors have addressed my previous comments with reasonable revisions. The additional smFRET datasets for nucleosomes alone and new FRET pair between H3 and H2A provide further support for the key findings of the work. The revised text places more emphasis on the novel findings and the added cartoons clarify the various pathways for dimer eviction.

The revised manuscript provides many new insights into the process of H2A.Z deposition. Most notably, the clear demonstration of the ATP-dependent steps and quantification of unique exchange pathways are nicely confirmed by the additional controls showing that unwrapping is driven by the eviction. In principle, I am supportive of the revised manuscript. However, upon carefully reviewing the revised work, I discovered several issues remaining that need to be addressed.

Comments:

1. The following citation should be included Poyton et al, Coordinated DNA and histone dynamics drive accurate histone H2A.Z exchange, *Sci. Adv.* 2022. Importantly, many of the results in this manuscript corroborate (and build upon) results from the paper above (insertion preference for distal linker, SR1C-mediated nucleosome un-wrapping events correlated to presence of ATP, etc.). Additionally, some of the experimental design demonstrates similarities (in vitro smFRET on +1-like nucleosomes...)

2. Fig 1

o I would expect the ET distribution at the start of the experiments (0 s) to be similar for all 4 plots because the exchange reactions were initiated by the addition of ATP. However, the second from the top plot in Fig 1d has a much different starting distribution than the other three plots. It is also hard to compare because the scales used for density and probability are not the same for all plots. Is it possible the wrong dataset is displayed in this panel? All the others seem to fit a trend. Could this be a different labeling position?

o In the experimental section, "The dimer eviction reaction was initiated 20 sec after beginning imaging by injecting 50 μ L reaction buffer containing the indicated concentration of ATP..." Fig1d would be less vague if the concentration of ATP used were indicated rather than the triangle representing [ATP].

o Caption says, "The eviction of H2A is marked by purple triangle." Should instead say rectangle.

o Caption should explicitly state what the "*" in Fig 1b represents.

o Figure 1A- it would be helpful to label the proximal and distal H2A/H2B dimers in the cartoon. Both the 117bp and 4bp ends of the DNA are referred to as "linkers" in the main text, so it is not immediately obvious which side is "linker-distal" or "linker-proximal". I would suggest removed all references to "linkers" in the text and only retaining the "linker-distal" or "linker-proximal" terms that are established in the field to avoid confusion.

o From the main text, "In addition, the rate of these transitions is dependent on ATP concentration (Figure 1D)". This would be more convincing if a quantitative measure of the rates were plotted at various [ATP], rather than histograms that demonstrate only the start and end ET distributions.

3. From the main text, "... the inclusion of nucleotide greatly increased the number of apparent dimer eviction events (ATP 0 μ M = 0.036 min⁻¹ ...". This is better described by frequency rather than number.

4. Fig 2

o From the main text, "Given that there are two H2A/H2B dimers per nucleosome, and the labeling of H2A was sub-stoichiometric, five types of eviction events are anticipated (Figure 2A-E) ... We also observed a population of nucleosome where the two dimers appeared to be evicted in a single step, though these events may also represent nucleosomes harboring only a single, labeled dimer (Figure 2E,F)." There seems to be quite a few more than five types of eviction events (such as single eviction of proximal dimer from dual-labeled nucleosome, which would be the opposite of D, or simultaneous

eviction as mentioned in the main text).

5. Fig S1

o Fig S1d y-axis label spelled incorrectly. It is unclear what "survival of release phase" refers to.

6. When priming is first brought up in the main text, it is unclear what may be occurring mechanistically because it looks like it is simply the amount of time until eviction occurs. This could simply be a lag before the nucleosome becomes associated with SWR1C and eviction occurs. It would be helpful if you could clarify what is meant by "priming" by including a sentence indicating what the molecular events are that would result in the lag seen.

7. Fig 3

o Caption should explicitly state what the "*" in Fig 3a represents.

o The number of * representing p-values should correlate with the p-value. In other words, ** for $p < 0.004$ and *** for $p < 0.0014$.

8. Fig 4

o Because frequency of unwrapping is being discussed, it would be more clear to mark the unwrapping events on the zoomed out plots in 4a since they have the same x-scale.

o Fig 4a: Center the zoomed-out plots over their corresponding zoomed-in plots

9. Fig S2

o Use "μM" instead of "uM"

o If this figure is meant to show "a significant increase in reversible FRET fluctuations... during the lag phase..." it would help to show a zoom in of some of those fluctuations, similar to fig 4a with the nucleosome unwinding.

10. Fig S3

o Can the plots be transparent? It is hard to see the orange under the blue.

11. Fig S4

o Fig S4b caption says labels are "*" and "#", which do not match the figure itself.

o Main text states, "Trajectories of Cy3B-H2A.Z fluorescence at individual nucleosome showed association of single or at times two or more Cy3B-H2A.Z/H2B dimers (Supplementary Figure S4)". I don't understand how S4 shows association of two or more dimers.

o "...binding events (Supplementary Figure S4B)" on page 9- This should reference S4C.

o "... within 10 seconds (Supplementary Figure S4C)" on page 9- This should reference S4D.

12. Fig S5

o Fig S5 caption says labels are "*" and "#", which do not match the figure itself.

o Main text: "However, as many of the tethered nucleosomes showed multiple dimers of Cy3B-H2A.Z/H2B simultaneously associated (Supplementary Figure S5)". It would be very helpful to the reader to point out which of these traces correspond to the multiple dimer associated species. Some of the traces are very noisy, and they are hard to distinguish.

13. Fig 5

o Fig 5a legend- labels should match the ones used in 5b instead of "Anuc"

o It would be good to include the plots for all of the samples graphed in 5b, rather than just H2A nuc + ATP or apo.

o In main text, "Together, these data indicate the strength of SWR1C-nucleosome interactions is modulated during the ATP cycle and hydrolysis may function to release the enzyme following gH2A.Z deposition." A fluorescence polarization assay using ADP.BeF3- could help shed light on this because ADP.BeF3- can function as a ATP hydrolysis transition state mimic.

Minor general comments

14. Panel labels ("A", "B", etc.) in figures are not aligned vertically nor horizontally, and 2F is the only label with a "." after it.

15. "One possibility is that the crosslinking assay is unable to capture changes in DNA trajectories due to a technical limitation." Huh. This is kind of vague.

16. "At saturating levels of ATP, the half-life for Cy3B-H2A release..." Did I miss where they determined saturating levels of ATP?

17. "These data indicate a weakened interaction between SWR1C and nucleosome post ATP hydrolysis..." Should say SWR1C instead of SWRIC. Also, couldn't this indicate weakened interaction during or post hydrolysis?

We thank the reviewers for their insightful and critical comments on our manuscript. In the revised manuscript, we had made all of the recommended minor changes to the text and figures, and we have added sentences throughout the manuscript that should provide extra clarification. Note that reviewer numbers have been changed to coincide with the first review at *NSMB*. Below, we comment on each reviewer point.

Reviewer 1

One of the findings of this paper regards what is called the priming phase which is a longer period of time that occurs before there is loss of FRET signal, presumably due to the loss of an H2A-H2B dimer. The authors conclude there is transient unwrapping of DNA from the histone octamer on the linker distal side in conjunction with destabilization of the histone octamer. The data however is not clear in this regard. In this revised version of the manuscript the authors have added new data by place the donor dye on histone H3 instead of on DNA and the acceptor dye is on histone H2A and then finding transient fluctuations like that observed when the donor dye is on DNA. They conclude that there are transient disruptions within the histone octamer that accounts for these transient fluctuations; however, they continue to state the other version with the donor dye on DNA shows that DNA is unwrapped. The original reason for performing this last experiment was to confirm if the fluctuations observed with the DNA-H2A FRET pair is truly a reflection of DNA unwrapping or could be the result of other structural fluctuations. It seems from their most recent experiment the conclusion should be they cannot state the changes in FRET signal with the labeled DNA and H2A are due to DNA unwrapping but instead are likely caused by changes in the histone octamer. If in the situation of the donor on DNA and acceptor on histone H2A, H2A is moving due to transient conformations of the histone octamer that would be sufficient to explain the type of fluctuations observed in this study and not due to the DNA moving in regard to the histone surface. In summary the authors don't have good evidence for DNA unwrapping from the linker distal side of the nucleosome and do not have an explanation for how ATP helps "prime" for eviction of the H2A-H2B dimer.

In response to the reviewer's previous concerns, we performed smFRET studies with a nucleosomal substrate containing FRET probes on H3 and H2A. Our primary objective was to prove that the stable transition to a low FRET state reflected H2A eviction and not stable DNA unwrapping. Indeed, this new substrate showed the same rapid (2-3") transition from a high FRET to low FRET state that followed a long period in the high FRET state, which we describe as the priming phase. Thus, this experiment confirms that we are monitoring eviction of H2A. We also found during the priming phase (high FRET state) that this substrate also exhibited transient FRET fluctuations. However, in contrast to the substrate harboring a fluor on the DNA end, in this case FRET fluctuations required both SWR1C and ATP. We also note in the revised version that recent work from the Wu group (Poyton et al., 2022) have used a three-color FRET approach to show that SWR1C induces transient changes in DNA-histone interactions. In general, we agree that the priming phase involves an overall destabilization of the nucleosome, involving transient changes in both DNA-histone interactions and histone-histone interactions. We make a reasonable conclusion that destabilization of the nucleosome will lower the energetic barrier for H2A eviction and replacement. We have revised the text to make this point clear and to not over-emphasize one or the other of these disruptions. (Page 15, top. "These data suggest that a model in which the long priming phase reflects the ability of SWR1C to destabilize the nucleosome by inducing both a transient unwrapping of DNA as well as deformations in the

H2A/H2B-H3/H4 interface, ‘priming’ the nucleosome and lowering the energetic barrier to the subsequent eviction and replacement of the H2A/H2B dimer.”)

In the discussion section they also mention they don't think these fluctuations are due to SWR1 binding because SWR1C is not influenced by ATP (see top of page 14). I found this statement to be contradictory to some of the new data that has been added as they now have fluorescence polarization data where they determine the affinity of SWR1 to H2A-nucleosomes with (1 mM) and without ATP, which showed that ATP decreased the affinity of SWR1 for H2A-nucleosomes from 12 to 31 nM and they conclude the strength of the SWR1C-nucleosomes is modulated with ATP. It seems therefore the fluctuations could be due to the dynamic binding and release of SWR1 from nucleosomes which would be influenced by ATP concentration, which further cloud the purpose of the long priming period and could it be merely that it has to keep on going through rounds of binding and releasing before it finally is successful for displacing H2A-H2B.

First, the FP data was not new to this revision, but was included in the previous version of this manuscript. Second, our binding data shows that ATP weakens binding of SWR1C, and thus the simple model would predict that increasing ATP would lengthen the priming phase, not decrease its length. Furthermore, the concentration of SWR1C was in large excess of nucleosomes, essentially single turnover conditions. We have added a sentence in the text to clarify these points. (Page 7, top “This phase is unlikely to reflect a lag in SWR1C nucleosome binding, as reactions were preincubated with SWR1C at concentrations promoting single turnover kinetics.)

A minor point is that although the affinity of SWR1 for H2A nucleosomes was measured w/o nucleotide (Apo) and with ADP, AMP-PNP and ATP, the affinity for H2A.Z nucleosomes was only measured w/o nucleotide. It seems like it would be critical to measure SWR1 affinity for H2A.Z nucleosome with ATP, ADP, and AMP-PNP as well.

The H2A.Z binding studies were added in response to previous reviewer request, and the primary goal was to demonstrate that SWR1C has a lower affinity for the product of the reaction. We anticipate that nucleotides are unlikely to change binding affinity, as an H2A.Z nucleosome does not promote SWR1C ATPase activity. Please see Luk et al., 2010.

Another key point of this paper is that SWR1C does not translocate on DNA based on site directed histone-DNA crosslinking which is compared to CHD1 and ISW2, two enzymes that dotranslocate when using the same assay. The model put forward by the authors is that SWR1C distorts DNA only at SHL2 without altering the register of DNA-histone contacts between the entry site and SHL2. They suggest the minor distortion in DNA coupled with distortion of the histone octamer would be the driving forces for dimer exchange. There are however concerns that the data is being misinterpreted.

The reviewer questions the experimental design and interpretation of our histone-DNA crosslinking studies. We agree with the reviewer that Chd1, Isw2, and Swr1 ATPases each interact with SHL^{+/-}2.0 and track on the same strand in a 3' to 5' direction. However, the cartoon and description provided by the reviewer are incorrect in several ways. First, our parallel analyses of Chd1, Isw2, and SWR1C used a center-positioned 40N40 nucleosome, not the end-positioned nucleosome shown in their diagram. The 40N40 is the identical substrate used in the

original Bowman work on Chd1 (Winger et al., 2018). On this substrate, all enzymes have the ability to interact with either SHL+2.0 or SHL-2.0. Previous work from Bowman demonstrated that Chd1 only alters DNA-histone crosslinks when bound to SHL+2.0, due to the sequence of the 601 element. This leads to the observed, new crosslinks at +55 (our work and work shown in Winger et al., 2018). Our data also indicates that ISW2 can induce new crosslinks when bound to either SHL-2.0 or SHL+2.0, yielding new crosslinks at -55 and +55. Notably, addition of SWR1C to this same nucleosome did not alter crosslinking.

Following analysis of the 40N40 substrate, we then asked whether SWR1C induced new crosslinks on the end-positioned nucleosome. On this substrate, Swr1 would interact with the bottom strand in the reviewer's cartoon, not the top strand, tracking 3' to 5'. The reviewer is correct that Isw2 and Chd1 would likely only interact with the opposite face, as these enzymes prefer to center nucleosomes. Consequently, we did not perform assays with these enzymes on the end-positioned substrate. The text has been revised to make sure each of these points are clear.

I didn't understand their suggestions of how Arp6/Swc6/Swc3 binding near the nucleosomal edge on the long linker side and the small distortion at SHL2 on the opposite side of the dyad axis, some ~90 or more nucleotides from each other, could work together to destabilize histone-DNA interactions without causing DNA to translocate as well. This model just doesn't make sense.

We apologize if our discussion was not clear. The model is that Arp6/Swc6/Swc3 interacts with the linker-distal nucleosome edge. This is on the same face as the Swr1 ATPase. We have now adjusted the text to make this more clear (Page 17, top. "SWR1C has a subunit module (Arp6/Swc6/Swc3) that binds to DNA at the linker-distal nucleosomal edge and may prevent the 'pulling' of DNA into the nucleosome")

What is missing in this study is an orthogonal approach to track loss of H2A-H2B dimer to validate their changes in FRET signal. Loss of FRET signal could indicate several types of events particularly since there is evidence the histone octamer is being deformed based on their H3-H2A FRET results. The orientation of the acceptor would need to merely change its location without being dissociated to produce the same change of FRET signal. Because of this lack of further validation it makes their interpretation of these data rather tenuous.

It is not clear what type of single molecule, orthogonal approach is suggested. We would point out that nucleosomal H2A harbors the donor fluorophore. We not only observe a stable loss of FRET with SWR1C and ATP, but we also see loss of the donor from the immobilized nucleosome (the release stage). These events are not due to photobleaching, and thus they must represent H2A eviction. Note that our previous ensemble work with this identical approach also compared rates of FRET loss to dimer exchange that was followed by a gel-based assay. Here, the rates were comparable, further validating the approach.

Minor points:

On page 6 the authors allude to the number of dimer eviction events being greater with ATP but don't tell you the number – instead they give you the rate differences. I would like to also

know the number of events as stated.

On page 6, we report the frequency of events. This has been corrected in the text. The actual number of events is shown in Table S1.

I see in Table S3 that the number of transient FRET fluctuations with and without ATP are only modestly different from each other (152 vs 234). It raises the question if the roughly doubling of the rate with the addition of ATP suggest these maybe intrinsic fluctuations that are modestly modulated by ATP and have no relationship to the process of dimer exchange?

While a 1.54-fold increase in the number of events may be modest, as the reviewer notes this leads to roughly doubling in rate of transient FRET fluctuations. In addition, these changes are clearly not intrinsic fluctuations, as they require SWR1C binding, and they are increased by ATP. Note as well that the dwell time of these low FRET events are also influenced by ATP.

When I look at what is referred to as the release phase in Figure 3, I'm not clear as to the significance of the release phase as not much really seems to be happening based on ET differences especially since ET is already so low and the donor intensity is so high. Is it really an important phase in the dimer exchange process and where is the rationale for it being so?

The release phase is defined as the period following the stable loss of FRET to when the H2A-donor signal is lost from the immobilized nucleosome (acceptor). Thus, during this period, ET is low due to H2A eviction, but the dimer (containing the donor fluor) remains associated with the SWR1C-nucleosome complex for an extended period. This release phase does seem to be important, as its duration is sensitive to ATP concentration.

The new results with labeled H3 and H2A are suppose to show the same 3 phases as seen before but in Supplemental Figure 2B-E I only see evidence for the eviction and priming phases. Where is the missing data? Is the lag period the same as the priming phase? Why is the fluctuations in the priming phase displayed so differently than before, making it difficult to compare and contrast the two? The data is also quite sparse here and begs the question as to the rigor of the experiment.

We apologize to the reviewer that the representative trajectory had not been labelled to show all three phases of the reaction. Note that the quantification of these three phases was shown in Table S2. We have also expanded the priming phase in a new Supplementary Figure S3 to illustrate the FRET fluctuations. We analyzed 323 events for this double-labelled histone substrate which we believe to be a highly significant number (Table S2).

Where is Supplementary Figure S4 – movie?

We apologize for the jargon; we were referring here to the acquisition period, not an actual movie. This sentence has been clarified as follows: Page 9, top "...at least one Cy3B-H2A.Z dimer over the course of 5-minutes of data acquisition".

There were only 29 deposition events observed out of a total of 1136 events. That number

makes one worried about what is being observed and if the system is adequate or optimal for tracking the incorporation of H2A.Z dimers into nucleosomes?

We apologize if the text was unclear. The 29 deposition events were observed for 309 nucleosomes where H2A.Z was co-localized, so nearly 10% efficiency. The 1136 events reflect all H2A.Z molecules that localize to nucleosomes. The text has been modified to make this clear.

I would like another orthogonal dimer incorporation assay before to support these data. The authors state the low number could be due to the incomplete labeling of H2A.Z. My question is if 100% of the H2A.Z dimer was labeled as compared to say ~80% do you think it would make that much of a difference?

The dimer deposition assay is technically challenging, as the labelled H2A.Z dimers tend to bind nonspecifically to the smFRET slide and to nucleosomes. The reviewer is correct that H2A.Z labelling efficiency is unlikely to be the sole cause of the low efficiency of this reaction. We have added the following additional sentence to the text: Page 9, bottom. "...photobleaching preceding deposition of H2A.Z, or a confounding impact of nonspecific H2A.Z binding."

What was the ATP concentration used for the H2A.Z deposition Sm FRET experiments? I could find it in either the figure legend, main text or methods section?

The ATP concentration was shown in the methods section (Page 29; 200 micromolar ATP)

The authors cite the early work with ACF using smFRET but have failed to reference later studies involving ISW2 where there do observe a similar lag before DNA is moved through the nucleosome that is not dependent on the kinetics of ISW2 and could be considered to resemble the data reported here (Deindl et al Cell 2013).

We did not cite work from Deindl and colleagues, as their experiments are not analogous to our assays. In this work, the authors monitored DNA movements both at the entry and exit sides of the nucleosome during an Isw2 remodeling reaction. They observed that Isw2 action led to movement of 7bp of DNA from the exit side prior to seeing DNA move into the nucleosome at the entry side. Consequently, there was a "lag" only on the entry side. There was no apparent "lag" on the exit side. Furthermore, the authors did not vary Isw2 concentration, so it remains unclear if the lag might be influenced by Isw2 binding kinetics. This was clearly demonstrated in the ACF paper that we cited.

Reviewer 2

We thank this reviewer for the careful editing of our manuscript, as well as their positive comments: "The additional smFRET datasets for nucleosomes alone and new FRET pair between H3 and H2A provide further support for the key findings of the work" and "the clear demonstration of the ATP-dependent steps and quantification of unique exchange pathways are nicely confirmed by the additional controls showing that unwrapping is driven by the eviction."

Comments:

1. The following citation should be included Poyton et al, Coordinated DNA and histone dynamics drive accurate histone H2A.Z exchange, *Sci. Adv.* 2022.

This citation has been added in several places (Page 8, top; Page 15, top).

2. Fig 1

I would expect the ET distribution at the start of the experiments (0 s) to be similar for all 4 plots because the exchange reactions were initiated by the addition of ATP. However, the second from the top plot in Fig 1d has a much different starting distribution than the other three plots. It is also hard to compare because the scales used for density and probability are not the same for all plots. Is it possible the wrong dataset is displayed in this panel? All the others seem to fit a trend. Could this be a different labeling position?

Unfortunately, the substrate used for this panel was under-labelled, compared to the other panels. Consequently, the starting distribution is skewed towards lower FRET values. Importantly, this does not impact the analysis of individual trajectories. This has now been noted in the Figure legend.

In the experimental section, “The dimer eviction reaction was initiated 20 sec after beginning imaging by injecting 50 μ L reaction buffer containing the indicated concentration of ATP...” Fig1d would be less vague if the concentration of ATP used were indicated rather than the triangle representing [ATP].

We would like to keep the triangle, as adding the individual labels seems to make the figure cluttered. The ATP values are stated in the legend.

Caption says, “The eviction of H2A is marked by purple triangle.” Should instead say rectangle. Caption should explicitly state what the “” in Fig 1b represents.*

Corrected and * defined.

Figure 1A- it would be helpful to label the proximal and distal H2A/H2B dimers in the cartoon. Both the 117bp and 4bp ends of the DNA are referred to as “linkers” in the main text, so it is not immediately obvious which side is “linker-distal” or “linker-proximal”. I would suggest removed all references to “linkers” in the text and only retaining the “linker-distal” or “linker-proximal” terms that are established in the field to avoid confusion.

The figure has been revised as suggested. We have clarified the text with respect to linkers, eliminating the statement referring to the 4bp linker.

From the main text, “In addition, the rate of these transitions is dependent on ATP concentration (Figure 1D)”. This would be more convincing if a quantitative measure of the rates were plotted at various [ATP], rather than histograms that demonstrate only the start and end ET distributions.

Our primary point was to conclude that the transition from high FRET states to low FRET states proceeded faster with increasing ATP. Since individual rates for transitions among different states are detailed in Table S1 and discussed subsequently, we have altered the text to state: (Page 6, first paragraph) “In addition, the transition to the low FRET population occurred more rapidly as the ATP concentration was increased (Figure 1D). Eviction events were defined as persistent loss of FRET from an initial state greater than $E.T. = 0.4$ to a final FRET state less than $E.T. = 0.35$. While loss of FRET was sometimes observed in the absence of ATP, these events are likely due to photobleaching, and the inclusion of nucleotide greatly increased the frequency of apparent dimer eviction events (ATP 0 $\mu\text{M} = 0.036 \text{ min}^{-1}$, ATP 0.5 $\mu\text{M} = 0.192 \text{ min}^{-1}$, ATP 5 $\mu\text{M} = 0.326 \text{ min}^{-1}$, ATP 100 $\mu\text{M} = 0.48 \text{ min}^{-1}$; Table S1).”.

3. From the main text, “... the inclusion of nucleotide greatly increased the number of apparent dimer eviction events (ATP 0 $\mu\text{M} = 0.036 \text{ min}^{-1}$...”. This is better described by frequency rather than number.

We apologize. We are reporting the frequency. We have changed the word “number” to “frequency”. (Page 6, “inclusion of nucleotide greatly increased the frequency of apparent dimer eviction events”)

4. Fig 2

From the main text, “Given that there are two H2A/H2B dimers per nucleosome, and the labeling of H2A was sub-stoichiometric, five types of eviction events are anticipated (Figure 2A-E) ... We also observed a population of nucleosome where the two dimers appeared to be evicted in a single step, though these events may also represent nucleosomes harboring only a single, labeled dimer (Figure 2E,F).” There seems to be quite a few more than five types of eviction events (such as single eviction of proximal dimer from dual-labeled nucleosome, which would be the opposite of D, or simultaneous eviction as mentioned in the main text).

The reviewer raises a valid point. We added the caveat “five types of primary events”, as these represent the vast majority of observed events. Eviction of the proximal dimer from the double labelled nucleosome is quite rare.

5. Fig S1

Fig S1d y-axis label spelled incorrectly. It is unclear what “survival of release phase” refers to.

Thank you, we have corrected the typo and have updated the figure legend to include “(D) Release phase survival kinetics (Kaplan-Meier estimate) for 100 μM ATP (N=431) (blue), 5 μM ATP (N=398) (purple) 0.5 μM ATP (N=227) (magenta). The x axis indicates survival time for the fraction of molecules yet to dissociate from SWR1C-nucleosome complex (solid line). Shaded areas, 95% confidence intervals.”

6. When priming is first brought up in the main text, it is unclear what may be occurring mechanistically because it looks like it is simply the amount of time until eviction occurs. This could simply be a lag before the nucleosome becomes associated with SWR1C and eviction occurs. It would be helpful if you could clarify what is meant by “priming” by including a sentence indicating what the molecular events are that would result in the lag seen.

The following sentence has been added: (Page 7, top. “Following addition of ATP, nucleosomes exhibited a “priming” period prior to a stable loss of FRET. This phase is unlikely to reflect a lag in SWR1C nucleosome binding, as reactions were preincubated with SWR1C at concentrations that should lead to single turnover kinetics.”

7. Fig 3

Caption should explicitly state what the “” in Fig 3a represents.*

*The number of * representing p-values should correlate with the p-value. In other words, ** for $p < 0.004$ and *** for $p < 0.0014$.*

Done

8. Fig 4

Because frequency of unwrapping is being discussed, it would be more clear to mark the unwrapping events on the zoomed out plots in 4a since they have the same x-scale.

Done

Fig 4a: Center the zoomed-out plots over their corresponding zoomed-in plots

Done

9. Fig S2

Use “ μ M” instead of “uM”

Done

If this figure is meant to show “a significant increase in reversible FRET fluctuations... during the lag phase...” it would help to show a zoom in of some of those fluctuations, similar to fig 4a with the nucleosome unwinding.

We have added a new Supplementary Figure S3 that shows the zoom-in trajectories.

10. Fig S3

Can the plots be transparent? It is hard to see the orange under the blue.

Rather than make the plots transparent, we have separated the datasets, as was done in Figure 4B.

11. Fig S4

Fig S4b caption says labels are “” and “#”, which do not match the figure itself.*

Corrected

Main text states, “Trajectories of Cy3B-H2A.Z fluorescence at individual nucleosome showed

association of single or at times two or more Cy3B-H2A.Z/H2B dimers (Supplementary Figure S4)". I don't understand how S4 shows association of two or more dimers.

We apologize, this should have been referring to Figure S5.

"...binding events (Supplementary Figure S4B)" on page 9- This should reference S4C.

Done

"... within 10 seconds (Supplementary Figure S4C)" on page 9- This should reference S4D.

Done

12. Fig S5

Fig S5 caption says labels are "" and "#", which do not match the figure itself.*

Corrected

Main text: "However, as many of the tethered nucleosomes showed multiple dimers of Cy3B-H2A.Z/H2B simultaneously associated (Supplementary Figure S5)". It would be very helpful to the reader to point out which of these traces correspond to the multiple dimer associated species. Some of the traces are very noisy, and they are hard to distinguish.

We now use a dotted line to denote dimer associations on Figure S5.

13. Fig 5

Fig 5a legend- labels should match the ones used in 5b instead of "Anuc"

We have changed these labels to H2Anuc

It would be good to include the plots for all of the samples graphed in 5b, rather than just H2Anuc + ATP or apo.

We have assembled a new Supplemental Figure S6 that shows all FP plots.

In main text, "Together, these data indicate the strength of SWR1C-nucleosome interactions is modulated during the ATP cycle and hydrolysis may function to release the enzyme following gH2A.Z deposition." A fluorescence polarization assay using ADP.BeF3- could help shed light on this because ADP.BeF3- can function as a ATP hydrolysis transition state mimic.

We have performed a limited number of FP assays with ADP-BeF3, and the data were quite similar to that obtained with AMP-PNP. However, we do not have as many biological replicas with this nucleotide state. It is not clear that this nucleotide analog does function as a transition state mimic for SWR1C, and given its little apparent impact, we would prefer to not include these data.

Minor general comments

14. Panel labels (“A”, “B”, etc.) in figures are not aligned vertically nor horizontally, and 2F is the only label with a “.” After it.

We have corrected much of these mis-aligned panels, though in some instances the alignment was due to differing panel sizes.

15. “One possibility is that the crosslinking assay is unable to capture changes in DNA trajectories due to a technical limitation.” Huh. This is kind of vague.

We have revised to: “One possibility is that the crosslinking assay is unable to capture changes in DNA trajectories due to a technical problem. We think this scenario is unlikely, given that the crosslinking assay is able to detect changes in DNA-histone interactions due to the ISW2 and Chd1 remodelers. Alternatively, SWR1-dependent changes in the path of nucleosomal DNA may be too rapid or dynamic to capture in the assay. We favor a model...”

16. “At saturating levels of ATP, the half-life for Cy3B-H2A release...” Did I miss where they determined saturating levels of ATP?

We now include a reference to earlier work from the Wu group who determined the K_m for ATP to be ~5 micromolar (Luk et al., 2010).

“At saturating levels of ATP (100 μ M; see Luk et al., 2010), the half-life for Cy3B-H2A...”

17. “These data indicate a weakened interaction between SWRIC and nucleosome post ATP hydrolysis...” Should say SWRIC instead of SWRIC. Also, couldn’t this indicate weakened interaction during or post hydrolysis?

Corrected

REVIEWERS' COMMENTS

Reviewer #1 (Remarks to the Author):

I appreciate the response given by the authors and the clarification they have provided. The text that has been changed on the top of page 15 helps better clarify. I still have a question as to what is the evidence they present that clearly shows there is transient unwrapping of nucleosomal DNA during the priming phase. If this statement is mostly dependent on the data from the Wu group in Poyton et al., 2022 then they should state that more clearly.

I agree with the authors that the stable reduction of the FRET signal is likely due to the loss of the H2A dimer, but also realize it could be other conformation changes that do not reflect complete loss of the H2A-H2B dimer. It is for this reason that it would be beneficial if there were other assays to further substantiate their FRET data.

I believe the authors have missed the point about ATP influencing the dynamics of SWR1. It is not about a lag in SWR1C binding to nucleosomes or whether nucleosomes are saturated by SWR1C, but instead if changes in FRET signal during the priming phase could be SWR1C mediated due to dynamic off and on rates of SWR1C that could be influenced by ATP concentration. This question remains unanswered by the authors or even considered in their discussion of the data. I don't think it would interfere with their desired final conclusions if they mention alternative interpretations. Increasing the off rate of ATP does not necessarily equate to decrease the efficiency of dimer exchange which would increase the length of the priming phase. Instead, it is possible by increasing the on and off rates and promoting more sampling of nucleosomes by SWR1C to enhance the rate of dimer exchange and thereby shorten the priming phase.

I think I need to explain why there is still doubt about the interpretation of the data and the authors' response regarding the histone-DNA crosslinking for measuring DNA translocation. When an asymmetric DNA is used for CHD1 versus SWR1C these two complexes engage nucleosomes quite differently and is the starting point for why it is challenging to do the head-to-head comparison. The Poyton et al. 2022 paper and others have shown that when using 80N0 that SWR1C initially is exchanging dimer at the linker distal position; whereas CHD1 and ISW1 start remodeling at the other side of the dyad axis with the same substrate. These data suggest that CHD1 and ISW2 likely use a different strand as the tracking strand than SWR1C. Next, the Winger et al. 2018 shows first that DNA movement can only be detected on one strand and not the other, which supports my concern about strand specificity being an issue. It is also important to point out that although CHD1 can bind to either SHL-2 or SHL+2 that DNA translocation can only be detected at the SHL+2 position and not at SHL-2, thus highlighting the nuances of this approach for tracking DNA translocation. Next, the movement shown in Figures 6 and S7 shows movement for ISW2 and CHD1 that is the same with or without nucleotide and doesn't really provide a good positive control for ATP-dependent DNA translocation. Probably the most relevant construct for SWR1C is likely the 77N4 given it most closely reflects the chromatin structure encountered at the promoter and is a more efficient substrate for dimer exchange than centered nucleosomes with 40-43 bp of linker DNA as shown in Poyton et al. 2022.

What is the evidence for Arp6/Swc6/Swc3 interacting with the linker distal side of the nucleosome? Would you still expect this module to bind at the linker distal side if there is only 4 bp of linker DNA and if these interactions were important for SWR1C remodeling why then can SWR1C remodel 80N0 nucleosomes like that in the Poyton et al. 2022 paper more efficiently than 40N40 nucleosomes?

Reviewer #2 (Remarks to the Author):

The authors have addressed most of my comments. I highlighted the merits of the study in my last review.

There were, however, a few comments that seem to have been missed. So I include those here for completeness.

Comment 2: We mentioned that it would be easier to compare the panels in Fig 1D if the scales for density and probability were the same for all four panels, but this was not addressed. This would also help the reader to clearly see "the transition to the low FRET population occurred more rapidly as the ATP concentration was increased (Figure 1D)" as the authors state.

Comment 7: I am still unsure what is going on with the p-values. Can't the number of * correlate with the magnitude of the p-value without defining them differently for each panel?

Comment 8: The unwrapping events are still not marked on the zoomed out plots, which would make the frequency easier to compare because they have the same time-scale.

Comment 12 part 2: Original comment was "However, as many of the tethered nucleosomes showed multiple dimers of Cy3BH2A.Z/H2B simultaneously associated (Supplementary Figure S5)". It would be very helpful to the reader to point out which of these traces correspond to the multiple dimer associated species. Some of the traces are very noisy, and they are hard to distinguish."

If I understand correctly, the authors added a dotted line to show binding of H2A.Z. However, this still doesn't help me understand where multiple dimers are associated, rather than just single.

Response to Reviewers Comments:

Reviewer #1

I appreciate the response given by the authors and the clarification they have provided. The text that has been changed on the top of page 15 helps better clarify. I still have a question as to what is the evidence they present that clearly shows there is transient unwrapping of nucleosomal DNA during the priming phase. If this statement is mostly dependent on the data from the Wu group in Poyton et al., 2022 then they should state that more clearly.

At the bottom of page 14 and top of page 15, we detail several lines of evidence in support of a transient unwrapping of nucleosomal DNA during the priming phase. This includes our previous FCS-FRET studies, as well as the work from Poyton et al. (2022). Our smFRET studies show transient loss of FRET when the fluorophores are located on the DNA edge and the H2A C-terminus. The simplest explanation for the data is that this reflects transient unwrapping of DNA from the nucleosome. This is also consistent with the Wigley cryoEM structure where unwrapping is observed in the presence of ADP-BeF. We attempted to present a balanced set of conclusions throughout the results and discussion, stating that the data suggests that priming is likely to reflect a combination of both DNA unwrapping and nucleosome conformational changes.

I agree with the authors that the stable reduction of the FRET signal is likely due to the loss of the H2A dimer, but also realize it could be other conformation changes that do not reflect complete loss of the H2A-H2B dimer. It is for this reason that it would be beneficial if there were other assays to further substantiate their FRET data.

We do not fully understand the reviewer's concern. During the dimer exchange reaction, we do not simply observe loss of the FRET signal (the eviction step), but we also see loss of the labelled dimer from the immobilized nucleosome (the release step). This latter step is not due to photobleaching, and thus it can only reflect dissociation of the dimer. It is also not clear what additional single molecule approach could be used to substantiate our data.

I believe the authors have missed the point about ATP influencing the dynamics of SWR1. It is not about a lag in SWR1 binding to nucleosomes or whether nucleosomes are saturated by SWR1, but instead if changes in FRET signal during the priming phase could be SWR1 mediated due to dynamic off and on rates of SWR1 that could be influenced by ATP concentration. This question remains unanswered by the authors or even considered in their discussion of the data. I don't think it would interfere with their desired final conclusions if they mention alternative interpretations. Increasing the off rate of ATP does not necessarily equate to decrease the efficiency of dimer exchange which would increase the length of the priming phase. Instead, it is possible by increasing the on and off rates and promoting more sampling of nucleosomes by SWR1 to enhance the rate of dimer exchange and thereby shorten the priming phase

We thank the reviewer for clarifying their model. We have now added a sentence in the Discussion that raises this alternative view (page 18, top).

I think I need to explain why there is still doubt about the interpretation of the data and the authors' response regarding the histone-DNA crosslinking for measuring DNA translocation. When an asymmetric DNA is used for CHD1 versus SWR1 these two complexes engage nucleosomes quite differently and is the starting point for why it is challenging to do the head-to-head comparison. The Poyton et al. 2022 paper and others have shown that when using 80N0 that SWR1 initially is exchanging dimer at the linker distal position; whereas CHD1 and ISW1 start remodeling at the other side of the dyad axis with the same substrate. These data suggest that CHD1 and ISW2 likely use a different strand as the tracking

strand than SWR1C. Next, the Winger et al. 2018 shows first that DNA movement can only be detected on one strand and not the other, which supports my concern about strand specificity being an issue. It is also important to point out that although CHD1 can bind to either SHL-2 or SHL+2 that DNA translocation can only be detected at the SHL+2 position and not at SHL-2, thus highlighting the nuances of this approach for tracking DNA translocation. Next, the movement shown in Figures 6 and S7 shows movement for ISW2 and CHD1 that is the same with or without nucleotide and doesn't really provide a good positive control for ATP-dependent DNA translocation. Probably the most relevant construct for SWR1C is likely the 77N4 given it most closely reflects the chromatin structure encountered at the promoter and is a more efficient substrate for dimer exchange than centered nucleosomes with 40-43 bp of linker DNA as shown in Poyton et al. 2022.

We agree with the reviewer that comparing SWR1C to Chd1/ISw2 on an asymmetric nucleosome is problematic, as Chd1/ISw2 and SWR1C prefer to engage different SHL positions. This is why we compared these three enzymes on the symmetric 40N40 substrate (Fig. 6b). Notably, this is the same substrate used by Bowman and colleagues for Chd1, which is why it is the appropriate positive control for this assay. We have not used asymmetric nucleosomes for Chd1 or ISw2. On the 40N40 substrate, we can monitor DNA strand movement whether the remodeler binds to either SHL+2 or SHL-2. Furthermore, on this substrate, SWR1C has no preference, especially under the saturation conditions that we employed.

We are not clear why the reviewer believes that SWR1C may use a different tracking strand than Chd1 or ISw2 – does he/she mean that Chd1 tracks on the Watson strand and SWR1C on the Crick? Or are they referring to which SHL2 the enzyme is docked for initiating remodeling? The cryoEM structure clearly shows that SWR1C uses the same tracking strand as other remodelers. This is also consistent with the blockage of dimer exchange by ssDNA gaps; work from both our group (Singh et al., 2019) and the Wu group (Ranjan et al., 2015).

The reviewer states that the movement is the same for Chd1 and ISw2 with or without nucleotide, and thus these controls are not effective. This is not quite correct. First, the key observation is that binding of these remodelers induces the 2nt shift in position (Fig 6b, Supplementary Fig. 7a,b). This reproduces what was known for Chd1, and we show the first data for ISw2. No such shift is seen for SWR1C under any condition. Second, in the presence of nucleotide, we see a clear loss of the overall crosslinked signal for both Chd1 and ISw2 in the presence of AMP-PNP (but not ADP); and for Chd1, this is clearly not due to sample loss (see full gel Supplementary Fig. 8d).

Note that we have also compared SWR1C activity on asymmetric and symmetric nucleosomes (Singh et al., 2019). In an ensemble reaction under single turnover conditions, SWR1C is slightly **more** active on a 50N77 nucleosomal substrate, compared to a 0N77 substrate. But we agree that the 077N is more relevant for SWR1C, and this is why this substrate (or similar) is used throughout.

What is the evidence for Arp6/Swc6/Swc3 interacting with the linker distal side of the nucleosome? Would you still expect this module to bind at the linker distal side if there is only 4 bp of linker DNA and if these interactions were important for SWR1C remodeling why then can SWR1C remodel 80N0 nucleosomes like that in the Poyton et al. 2022 paper more efficiently than 40N40 nucleosomes?

In the cryoEM structure of the SWR1C-nucleosome complex (Wilhoft et al., 2018), the Arp6/Swc6/Swc3 module is bound to the DNA edge that is on the same nucleosomal face as the Swr1 ATPase. Therefore, in this structure, the H2A/H2B dimer to be evicted is flanked by these two sets of DNA contacts. Both our current study and our previous work confirms that SWR1C prefers to evict the linker distal dimer on these asymmetric nucleosome substrates, which is why we have placed the Arp6/Swc7/Swc3 module at this position in our model. We do not believe that the 4bp linker influences our model, as we find that SWR1C still prefers to evict the linker (77bp) distal dimer. As noted above, we reported previously that

SWR1C catalyzes dimer eviction slightly faster on a symmetric nucleosome under single turnover conditions (Singh et al., 2019).

Reviewer #2

The authors have addressed most of my comments. I highlighted the merits of the study in my last review.

There were, however, a few comments that seem to have been missed. So I include those here for completeness.

Comment 2: We mentioned that it would be easier to compare the panels in Fig 1D if the scales for density and probability were the same for all four panels, but this was not addressed. This would also help the reader to clearly see “the transition to the low FRET population occurred more rapidly as the ATP concentration was increased (Figure 1D)” as the authors state.

We apologize for not directly addressing this point, as the previous concern was centered on the 0.5 μ M ATP data which we explained was due to under-labeling of this substrate. The scale for density (left panels, Fig. 1d) are in fact the same between all left panels. The probability scales (right panels; Fig. 1d) are different for the (-)ATP and 0.5 μ M ATP conditions in order to illustrate the smaller number of events. Note that these histograms present start and end points of the reactions, and thus do not report on rates.

*Comment 7: I am still unsure what is going on with the p-values. Can't the number of * correlate with the magnitude of the p-value without defining them differently for each panel?*

In keeping with Nat Comm policy, we have now replaced the * with the p-value numbers.

Comment 8: The unwrapping events are still not marked on the zoomed out plots, which would make the frequency easier to compare because they have the same time-scale.

Perhaps we have mis-interpreted the original comment, but Fig. 4a does denote the unwrapping events with an * in the zoomed-out plots. The time-scales are somewhat different between -ATP and +ATP, as we wished to show multiple events, and they are less frequent in the absence of ATP.

Comment 12 part 2: Original comment was “However, as many of the tethered nucleosomes showed multiple dimers of Cy3BH2A.Z/H2B simultaneously associated (Supplementary Figure S5)”. It would be very helpful to the reader to point out which of these traces correspond to the multiple dimer associated species. Some of the traces are very noisy, and they are hard to distinguish.”

If I understand correctly, the authors added a dotted line to show binding of H2A.Z. However, this still doesn't help me understand where multiple dimers are associated, rather than just single.

To better demonstrate the binding of multiple H2A.Z dimers to the immobilized SWR1C-nucleosomes, we have generated a new Supplementary Fig. 6 which show example trajectories of total donor intensity (sum of donor emission and donor excited acceptor emission). Stepwise changes in total donor intensity indicate colocalization of multiple H2A.Z with the SWR1C-nucleosome complex.